# Characterization of dust‑related new particle formation events based on long‑term measurement in North China Plain

Xiaojing Shen[1], Junying Sun[1,2], Huizheng Che[1], Yangmei Zhang[1], Chunhong Zhou[1], Ke Gui[1], Wanyun Xu[1], Quan Liu[1], Junting Zhong[1], Can Xia[1,3], Xinyao Hu[1,4], Sinan Zhang[1,5], Jialing Wang[1], Shuo Liu[1], Jiayuan Lu[1], Aoyuan Yu[1,4], Xiaoye Zhang[1]

[1]State Key Laboratory of Severe Weather & Key Laboratory of Atmospheric Chemistry of CMA, Chinese Academy of Meteorological Sciences, Beijing, 100081, China.

[2]State Key Laboratory of Cryospheric Science, Northwest Institute of Eco-Environment and Resources, Chinese Academy of Sciences, Lanzhou, 730000, China.

[3]Nanjing University of Information Science & Technology, Nanjing, 210000, China

[4]University of Chinese Academy of Sciences, Beijing, 100049, China

[5]Shaanxi Meteorological Observatory, Xi'an, 710014, China

*Corresponding to:* X. J. Shen (shenxj@cma.gov.cn)

**Abstract.** Mineral dust is a major natural atmospheric aerosol that impacts the Earth's radiation balance. The significant scavenging process of fine particles by the strong wind during the dust provided a relatively clean environment that was favourable for new particle formation (NPF) occurrence. In this study, the NPF occurred following the dust event (dust-related NPF) and other cases under clean and polluted conditions were classified based on the long-term particle number size distribution (PNSD) in urban Beijing in spring from 2017 to 2021. It was found the observed formation ($J_{obs}$) and growth rate ($GR$) of dust-related NPF events were approximately 50% and 30% lower than the values of other NPF days, respectively. A typical severe dust storm that originated from Mongolia and swept over northern China on March 15–16, 2021 was analysed, to illustrate how the dust storm influences the NPF event. The maximum hourly mean $PM_{10}$ mass concentration reached 8000 μg m$^{-3}$ during this dust storm. The occurrence of NPF event after dust storm was facilitated due to the low condensation sink (~0.005 s$^{-1}$) caused by the strong dilution process of pre-existing particles. However, a downward trend of particle hygroscopicity was found during dust storm and NPF event as compared with the polluted episode, resulting in an increasing trend of the critical diameter at different supersaturations (*ss*) where aerosols are activated as cloud condensation nuclei (CCN), although NPF event occurred when dust faded. The critical diameter was elevated by approximately 6%–10% (ss = 0.2% and 0.7%) during the dust storm, resulting in a lower CCN activation ratio, especially at low supersaturation. Modifications of the nucleation and growth process, as well as the particle-size distribution and hygroscopicity by the dust, provide valuable information that reveals the underlying climate and air quality effects of Asian mineral dust.

**1 Introduction**

New particle formation (NPF) events have been identified as a major particle source and can produce approximately 50% of the cloud condensation nuclei (CCN), which leads to significant but poorly quantified radiative forcing (Gordon et al., 2017; Yu and Luo, 2009). Several studies have reported that NPF events can occur globally, including in pristine, urban, rural, forest, mountaintop, and coastal environments (Bianchi et al., 2016; Dada et al., 2017; Jokinen et al., 2018; Kulmala et al., 2004; Shen et al., 2011). Unlike relatively clean regions, the nucleation and subsequent growth processes are complex in polluted urban environments, due to the incomplete understanding of the dynamics of nano-particles and clusters under highly polluted conditions (Cai and Jiang, 2017; Kulmala et al., 2017). It also remains challenges to quantify the contribution of NPF events to haze formation in China because it is difficult to separate aerosols from primary sources and gas-to-particle formation (Kulmala et al., 2022).

Mineral dust particles are another important aerosol type in the atmosphere that primarily originate from arid and semiarid regions. They can significantly affect the radiative balance of the Earth's system by scattering and absorbing solar radiation, as well as the formation and properties of clouds by acting as CCN and ice nuclei (DeMott et al., 2010; Liao and Seinfeld, 1998; Seinfeld and Pandis, 1998; Twohy et al., 2009). Model simulations were performed with and without dust influence, and the results predicted that total particle concentration and CCN were reduced by approximately 20% and 10%, respectively, as influenced by the dust pollution plume in East Asia (Manktelow et al., 2010). Dust particles can also lead to radiative feedback in the planetary boundary layer and lift dust particles to higher altitudes (Liu et al., 2016). The heterogeneous reactions of mineral dust with trace gases in the atmosphere can alter the chemical and physical properties of aerosols, including particle hygroscopicity (Ge et al., 2015; Tang et al., 2017). Heterogeneous oxidation of $SO_2$ onto particles has been observed, and is an important mechanism for converting $SO_2$ into sulfate (Li et al., 2011). Several laboratory and field studies have focused on the formation of secondary aerosols on dust particles (Liu et al., 2013; Xu et al., 2020). Dust particles enhance the reactive surface areas, absorb trace gases (Ma et al., 2017), and can further modify the chemical composition of the particles. Previous studies have also shown that the secondary formation of inorganics on the dust surface can enhance solubility and hygroscopicity (Mori, 2003; Perry et al., 2004).

In the economic developed and densely populated North China Plain (NCP) region, aerosols are dominated by anthropogenic emissions, and can cause serious air pollution (Zhang et al., 2019). However, based on the optical parameters, including particle linear depolarization ratio, volume linear depolarization ratio and lidar ratio derived from a Raman lidar, there were approximately 45% of aerosols below 1.8 km above the ground contributed by polluted dust (the mixture of anthropogenic aerosols and dust) in Northern China (Wang et al., 2021). It has also been reported that the coarse mode (diameter $\geq 1$ μm) serves as a medium and promotes rapid secondary aerosol formation, driving severe haze formation in the NCP region of China (Xu et al., 2020). However, the impact of dust bursts on nucleation and growth processes based on long-term measurement has

not been discussed in urban areas in China based on open literature.

In this work, we analyzed NPF events based on long-term measurement of particle number size distribution (PNSD), to characterize NPF events influenced by dust. The occurrence of NPF events under the clean atmosphere with extremely low condensation sink (*CS*) when the dust events fade, which helps to evaluate how the dust events modify the atmospheric conditions and facilitate the nucleation and growth processes. Specifically, a case study of a typical NPF event occurring after a severe dust storm is discussed in detail. A severe sand and dust storm (SDS) hit North East Asia from March 15 to 16, 2021, sweeping from Mongolia, through most parts of North China, and the Korean Peninsula, causing widespread damage, severe air pollution, and low visibility. This dust storm has been reported to be the most intensive event over the last two decades based on satellite and ground-based observations (Gui et al., 2022). For this specific case, the influence of dust on the NPF event, including the number/volume size distribution, chemical composition, and hygroscopicity of submicron particles, as well as CCN-sized particles, was analyzed to reveal the underlying climate and air quality effects of a typical severe Asian mineral dust storm.

## 2    Methodology

### 2.1    Sampling site

The physical and chemical properties of the particles were measured on the roof of the Chinese Academy of Meteorological Sciences (CAMS) building on the Chinese Meteorological Administration campus. The site is located approximately 53 m above ground level in the western Beijing urban area between the second- and third- ring roads. A major road with heavy traffic to the west of the site indicates that the sampled air could be influenced by traffic emissions. More information on the site can be found in the following studies (Shen et al., 2019; Wang X. et al., 2018).

### 2.2    Instrumentation

Ambient aerosols were sampled through a $PM_{10}$ impactor with a total flow rate of 16.7 Lpm. Different aerosol instruments were applied, including a tandem scanning mobility particle sizer (TSMPS, TROPOS, Germany). TSMPS system consisting of two differential mobility analyzers (DMAs, TROPOS, Germany) and two condensation particle counters (CPCs, models 3772 and 3776, TSI Inc., St Paul, USA), were used to measure the particle number size distributions (PNSDs) of 3–850 nm in mobility diameter at the CAMS site from 2017 to 2021. In this study, as we focused on dust events concentrated in spring, the data from March, April, and May from 2017 to 2021 were analyzed. Due to the malfunction of CPC 3776, which measured the PNSDs below 40 nm, the data in spring 2020 were excluded as the formation and growth rate of NPF days could not be precisely identified. More details about PNSD measurement setup can refer to Shen et al. (2021).

During the extensive campaign in the spring of 2021, in addition to TSMPS, other instruments including an aerodynamic

particle sizer (APS, model 3321, TSI Inc., USA), hygroscopicity tandem differential mobility analyzer (H-TDMA, TROPOS, Germany), and an aerodyne high-resolution time-of-flight aerosol mass spectrometer (HR-ToF-AMS, Aerodyne Research, Inc., USA) shared a common inlet, and the relative humidity (RH) of the sample air was controlled below 30% with an automatic regenerating absorption aerosol dryer system.

Particles with an aerodynamic diameter in the size range of 0.5–10 μm were derived using APS. Combined with TSMPS data, the PNSD can be used to calculate the surface and volume concentrations, assuming a spherical particle shape. However, owing to the non-sphericity of the dust particles, the surface area cannot be directly converted. Previous studies have also revealed that APS can undersize particles with irregular shapes (Cheng et al., 1990) and oversize dense particles (Barron, 1996), and may result in approximately 10–30% under-sizing of dust particles (Cheng et al., 1990). The particle number concentration in the overlap size range of 500-850 nm derived from TSMPS and APS was compared and the bias was smaller depending on the increasing particle size. For the particles of ~850 nm, the ratio of number concentration from TSMPS to that from APS was about 1.5. The mean and standard deviation of PNSD and volume size distribution derived from TSMPS and APS was given in the supplementary materials, MS (Fig. S1).

The H-TDMA system is comprised of two DMAs, a CPC (Model 3772, TSI Inc., USA) and a humidifier system between the two DMAs. The first DMA selects the quasi-monodisperse particles at a diameter ($D_{p,dry}$= 50, 100 nm) under the dry state with 30% RH (Maβling et al., 2003). Then, the size-selected particles pass through a humidity conditioner, which can be adjusted to the setting RH of 90%. The probability distribution function (PDF) of hygroscopic growth factor (HGF), HGF-PDF is inverted by the TDMAinv method developed by Gysel et al. (2009).

The chemical composition of non-refractory $PM_1$, including organic components, sulfate, nitrate, ammonium, and chloride, was derived using HR-ToF-AMS with a 5-min resolution (Drewnick et al., 2005). The calibrations of ionization efficiency (IE) were performed, using size-selected (300 nm) ammonium nitrate particles before and after the experiment. Default relative IE values were used for organics (1.4), nitrate (1.1), sulfate (1.2), ammonium (4.0), and chloride (1.3), respectively. The HR-ToF-AMS collection efficiency (CE) accounts for the incomplete detection of aerosol species owing to particle bounce at the vaporizer, and/or the partial transmission of particles by the lens (Canagaratna et al., 2007). In this study, a composition-dependent CE correction was used, following the methodology described by Middlebrook et al. (2012). Positive matrix factorization (PMF) (Ulbrich et al., 2009) and a multilinear engine (ME-2) (Canonaco et al., 2013) modelling of high time resolution organic mass spectrometric data from HR-ToF-AMS have also been used to resolve organics into primary organic aerosols (POA) and oxygenated organic aerosols (OOA), which correspond to different sources and processes (Zhang et al., 2022).

The mass concentrations of $PM_{2.5}$ and $PM_{10}$ with hourly time resolution at the selected air quality monitoring sites were

derived from the China National Environment Monitoring Center (CNEMC, http://www.cnemc.cn, last access: October 25, 2022). Trace gases at the CAMS site were measured simultaneously using a set of online analyzers from the Thermo Electron Corporation (USA), including $SO_2$ (43 CTL), $O_3$ (49 C), and $NO_2$ (42 CTL). The TE 49 C has a lower detection limit of 1 ppb and a precision of 1 ppb. The 42 CTL has a lower detection limit of 50 ppt and a precision of 0.4 ppb. The 43 CTL has a lower detection limit of 0.1 ppb and a precision of 1 ppb (Lin et al., 2009). Measurement signals of trace gases were recorded as 1 min averages (Lin et al., 2011), however, the hourly average data were used for discussion, in order to match with the PM mass concentration data.

## 2.3 Calculations

### 2.3.1 NPF classification, formation and growth rate

The classification of NPF events was based on the principles and methods presented by Dal Maso et al. (2005), in which a distinct new mode of particles (3–25 nm) had to appear in the size distribution of the nucleation mode and grow into larger diameters in the following hours after nucleation started. The parameters characterizing NPF events, observed formation rate at 3 nm ($J_{obs}$), growth rate ($GR$), as the $CS$ can be determined by the PNSD measurement, as suggested by Kulmala et al., (2012).

$$J_{obs} = \frac{dN_{3-25}}{dt} + CoagS \times N_{3-25} + \frac{GR_{3-25}}{\Delta dp} \times N_{3-25} \qquad (1)$$

where $CoagS$ is the coagulation sink and $GR_{3-25}$ is the growth rate from 3 nm to 25 nm. Further, $GR$ is defined as the diameter rate of change with time: $GR = (D_{p,2} - D_{p,1})/dt$ (nm h$^{-1}$), where $D_{p,1}$ and $D_{p,2}$ are the geometric mean diameters ($D_{p,g}$) when the nucleated particles start and stop growing, respectively. $D_{p,g}$ can be derived using lognormal fitting algorithms (Hussein et al., 2005).

### 2.3.2 Oxidation ratio of secondary inorganics

The sulfur oxidation ratio (SOR) and nitrogen oxidation ratio (NOR) are often used to estimate secondary sulfate and nitrate formation via the reactions of $SO_2$ and $NO_2$ (Wu et al., 2020). SOR and NOR are defined as the molar ratios of sulfate and nitrate to total oxidized sulfur and nitrogen, respectively, and are calculated as follows:

$$SOR = \frac{n[SO_4^{2-}]}{n[SO_4^{2-}] + n[SO_2]} \qquad (2)$$

$$NOR = \frac{n[NO_3^-]}{n[NO_3^-] + n[NO_2]} \qquad (3)$$

where n represents the molar concentration.

### 2.3.3 Correction of the dust particle diameter

The TSMPS system gives the PNSD as a function of mobility diameter ($D_p$), which can represent the volume equivalent

diameter ($D_{p,ve}$) for the spherical particles. The aerodynamic diameter ($D_{p,a}$) given by APS can be converted to the mobility diameter by applying the particle density ($\rho$) and shape factor ($\chi$), which can be used to derive the entire particle size range (3–10 μm) (Hinds, 1999; Reid, 2003):

$$D_{p,ve} = D_{p,a} \sqrt{\chi \frac{1}{\rho}} \times \sqrt{\frac{C_c(D_{p,a})}{C_c(D_{p,ve})}} \tag{4}$$

where $C_c$ is the Cunningham slip correction factor, and $C_c(D_{p,a})$ nearly equals to $C_c(D_{p,ve})$ due to the particles derived by APS being in the continuum region. For bulk dust aerosols, a density of 2.5 g cm$^{-3}$ and a mean shape factor of 1.8, which ranged from 1.6 to 2.6, were applied in this study (Reid et al., 2008).

**2.3.4    Hygroscopicity parameter**

The hygroscopic parameter ($\kappa$) can be calculated using the approximate expression suggested by (Petters and Kreidenweis, 2007):

$$\kappa_{htdma} = (HGF^3 - 1) \left( \frac{\exp\left(\frac{A}{D_{p,dry} \times HGF}\right)}{RH} - 1 \right) \tag{5}$$

$$\kappa_{ccn} = \frac{4A^3}{27D_{p,crit}^3 \ln^2 S_c} \tag{6}$$

$$A = \frac{4\sigma_{s/a} M_w}{RT\rho_w} \tag{7}$$

where $D_{p,dry}$ (100 nm) and HGF are the initial dry particle diameter and hygroscopic growth factor at RH (90%) measured by H-TDMA, $\rho_w$ is the density of water (1.0 g cm$^{-3}$), $M_w$ is the molecular weight of water, $\sigma_{s/a}$ is the surface tension of the water solution (0.0728 N m$^{-2}$), $R$ is the universal gas constant, $T$ is the temperature, and $D_{p,crit}$ is the critical value at which 50% of the particles are activated at supersaturation ($S_c$).

# 3    Results and discussion

## 3.1 The overall dust-related NPF events from 2017 to 2021

The dust days were classified into three types based on visibility (National Weather Bureau of China, 1979; Wang et al., 2005), including dust storms with sand being lifted by strong winds and visibility (vis) < 1.0 km; blowing dust formed by strong wind carrying a lot of dust and sand, with visibility of 1.0-10 km, and floating dust with fine dust suspended in the lower troposphere with vis ≤ 10 km. In this study, the visibility data are from the national surface meteorological observation stations of China Meteorological Administration (CMA). Furthermore, the daily weather phenomena and visibility are issued by CMA (http://www.asdf-bj.net/publish/observation/5.html, last access on March 23, 2023), which can also help to identify the dust event. The potential source contribution function (PSCF) analysis showed (Fig. S2 in SM) that high PM$_{2.5}$ mass

concentration at CAMS was dominated by two sources, the north-westerly and westerly originating air mass containing dust particles, and the southerly air mass with high mass loading of anthropogenic aerosols. However, for $PM_{10}$ mass concentration, the high values only contributed by the air masses passing through Inner Mongolia and carrying dust particles.

From 2017 to 2021, dust storms (DS), blowing dust (BD), and floating dust (FD) occurred once on March 15 2021, 13 days, and 25 days, respectively. Following approximately 80% of the dust days, the NPF event subsequently occurred, as shown in Fig.1. Considering the total NPF days in each spring (March, April, May) from 2017 to 2021, there were 44 days in 2017, 29 days in 2018, 30 days in 2019 and 31 days in 2021. NPF events were classified into different types, which occurred under clean conditions (103 clean NPF cases), polluted conditions (10 polluted NPF cases) and events influenced by BD (9 cases), FD (11 cases) and DS (1 case), respectively. The clean and polluted NPF events were identified according to the *CS* criterion value of 0.027 $s^{-1}$, which was the median value during the measurement. The non-dust related NPF events with *CS* above this value was regarded as the polluted case, otherwise, the cases were the clean NPF events. The abrupt outbreak of the coronavirus disease 2019 (COVID-19) in early 2020 and the preventive restrictions of human activity resulted in the reduction of anthropogenic emissions (Le et al., 2020). Besides the instrument malfunction as mentioned above, in order to exclude the influence of the COVID-19, NPF events in the spring of 2020 were not discussed in this study. However, we elucidated that the elevated atmospheric oxidizing capacity during the lockdown period favored nucleation and growth processes in urban Beijing (Shen et al., 2021).

It was found the NPF events could be interrupted or prohibited by the prevailing of dust process or the mixture with anthropogenic emissions, which could explain the other 20% of the dust days without NPF events afterwards. NPF events usually occurred around 8:00-12:00 LT when solar radiation increased, so if the dust processes prevailed during this period, it could prevent NPF occurrence. For some cases of blowing and floating dust days overlaid by the anthropogenic aerosols, for example, March 18, 2019, April 5, 2019 and May 15, 2019, the *CS* remained high during the dust (above 0.027 $s^{-1}$), which was unfavorable for NPF occurrence. For some cases (May 6-8, 2021, Fig. S3 in SM), the whole dust process was composed of blowing and floating dust episode depending on the strength of air masses containing dust particles influenced Beijing. NPF event can not be observed until the whole dust process finished. On May 7 and 8, NPF events were observed with extremely low *CS* values of approximately 0.0025-0.003 $s^{-1}$, indicating the concentration level of precursors participating nucleation and growth were comparable for these two cases. Although there was no direct $H_2SO_4$ measurement available in this work, the proxy sulfuric acid concentration was estimated as recommended by Lu et al., (2019), and the details were given in SM. The proxy $H_2SO_4$, as well as $SO_2$ and $NO_2$ concentrations were similar on May 7 and 8 (Figs. S4, 5 in SM). Under specific conditions, for example, the nucleation process was observed on March 28, 2021, however, the growth process was interrupted by the elevated background aerosol concentration, indicated by the increasing *CS* (Fig. S6 in SM). In addition, the mixture of

dust and anthropogenic aerosols from southerly air mass could result in high *CS* and thus NPF event was prohibited (Figs. S7,

8 in SM).

Table 1. The date of different dust types observed in Beijing from 2017 to 2021

| Dust type | Date (yyyy-mm-dd) |
|---|---|
| Dust storm | 2021-03-15 |
| Blowing dust | 2017-05-04, 2017-05-05, 2017-05-11 |
| | 2018-05-28 |
| | 2019-3-29, 2019-4-18, 2019-11-17 |
| | 2020-03-18, 2020-05-11 |
| | 2021-03-28, 2021-04-15, 2021-05-06, 2021-05-07 |
| Floating dust | 2018-03-28, 2018-05-01, 2018-05-05 |
| | 2019-02-05, 2019-03-18, 2019-03-20, 2019-03-27, 2019-04-05, 2019-05-12, 2019-05-15, 2019-10-28 |
| | 2020-04-24, 2020-05-13, 2020-10-21 |
| | 2020-10-31, 2021-01-13, 2021-01-15, 2021-02-21, 2021-02-22, 2021-03-30, 2021-03-31, 2021-04-01, 2021-04-16, 2021-05-08, 2021-05-23 |

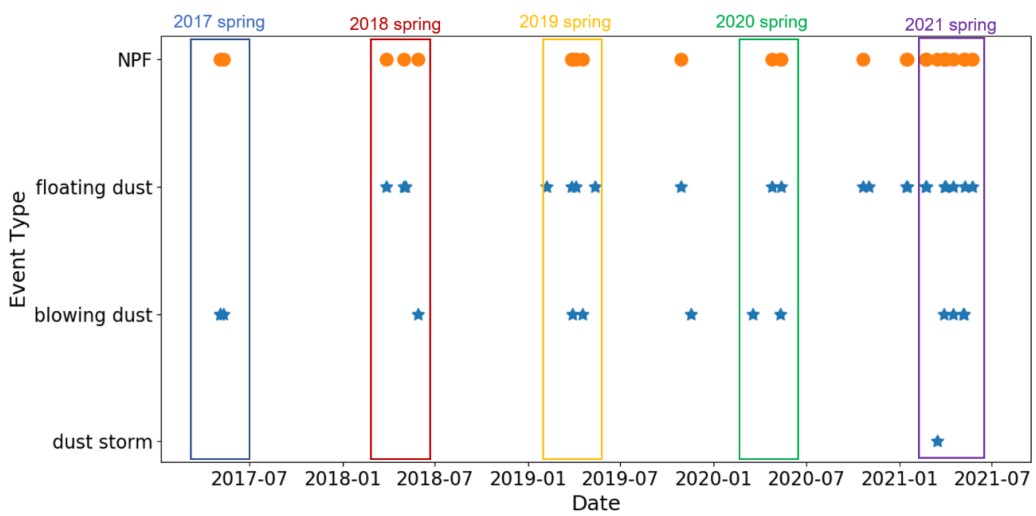

Fig.1 The occurrence of dust events, including dust storms, blowing dust, and floating dust from 2017 to 2021, and the NPF events observed after each dust event. The stars and dots represent the dust and NPF events, respectively.

The anomaly plots were obtained by means of the PNSD of NPF occurring on non-dust days subtracting the mean PNSD of dust-related NPF in each spring from 2017 to 2021, and was shown in Fig. 2. The positive anomaly indicated how much the particle number concentration in each corresponding size bin on non-dust NPF days was higher than that on dust-related NPF days, whereas the negative anomaly indicated that PNSD was lower on non-dust NPF days. Positive PNSD anomalies (Fig. 2a, c, e, g) in nucleation and Aitken mode showed stronger nucleation and growth processes. The calculated increase rates of

nucleation mode particles according to the positive anomaly of PNSD, can be considered as the enhancement of formation rate on non-dust NPF days. Table 2 lists the statistical values of dust-related NPF events and other NPF events. The observed formation rate on dust-related NPF days ($J_{obs,dust\_NPF}$) accounted for approximately 43%-58% of the value on other NPF days ($J_{obs,other\_NPF}$) from 2017 to 2021, with a mean value of approximately 51%. $GR$ of dust related-NPF ($GR_{dust\_NPF}$) ranged from 2.0 to 3.1 nm h$^{-1}$, whereas the value of other NPF days ($GR_{other\_NPF}$) was 3.6 to 4.4 nm h$^{-1}$. The ratio of $GR_{dust\_NPF}$ to $GR_{other\_NPF}$ ranged from 0.50 to 0.86, with a mean value of approximately 0.67. However, the growth process usually undergone several hours, with considerable anthropogenic emissions contributing to the particle growth.

Table 2. The statistical values of NPF in spring from 2017-2021, including the number ($N$) of dust related NPF events, other NPF events and observed formation rate ($J_{obs}$), and growth rate ($GR$) of these two types NPF events

|  | Spring 2017 | Spring 2018 | Spring 2019 | Spring 2021 |
|---|---|---|---|---|
| $N_{dust\_NPF}$ | 3 | 3 | 4 | 11 |
| $N_{other\_NPF}$ | 41 | 26 | 26 | 20 |
| $J_{obs,dust\_NPF}$ (cm$^{-3}$ s$^{-1}$) | 1.5±0.2 | 3.3±2.1 | 4.6±2.4 | 3.8±1.0 |
| $J_{obs,other\_NPF}$ (cm$^{-3}$ s$^{-1}$) | 3.3±2.6 | 7.3±4.8 | 8.1±4.5 | 6.7±3.2 |
| $GR_{dust\_NPF}$ (nm h$^{-1}$) | 2.0±0.6 | 2.9±0.3 | 3.1±0.5 | 2.8±1.2 |
| $GR_{other\_NPF}$ (nm h$^{-1}$) | 4.0±1.6 | 4.4±2.0 | 3.6±1.5 | 4.3±1.9 |

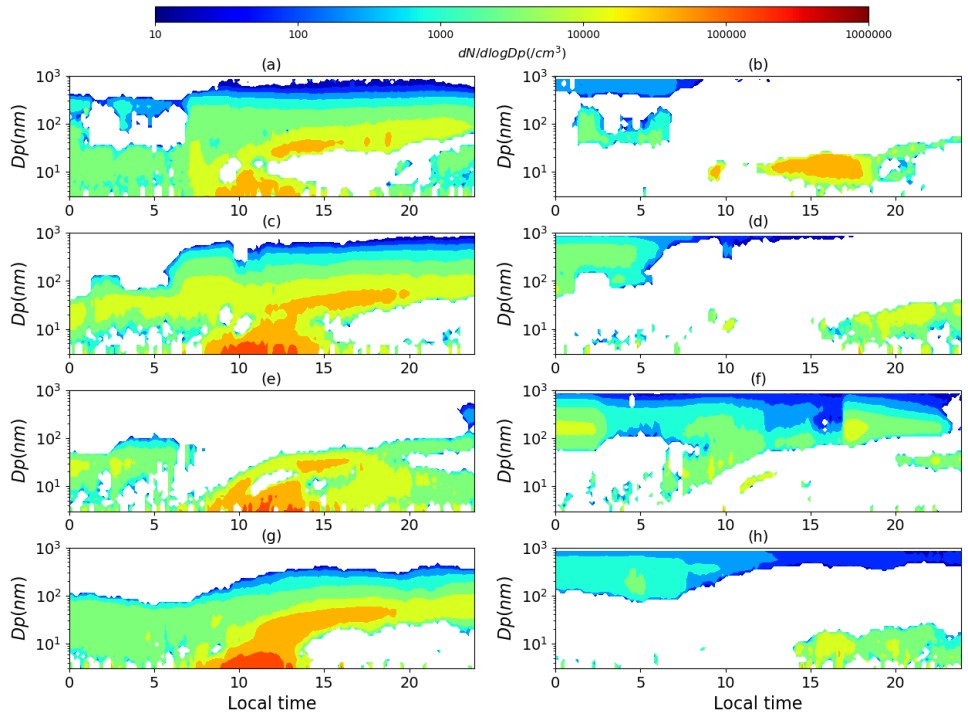

Fig. 2 The positive (a, c, e, g) and negative (b, d, f, h) anomalies of PNSD derived by the mean PNSDs of NPF events on non-dust days in spring (March, April, and May) subtracting the means of dust-related NPF days in 2017 (a, b), 2018 (c, d), 2019 (e, f) and 2021 (g, h).

The scatterplot of $J_{obs}$, $GR$ and $CS$ categorized by different NPF event types was given in Fig. 3. It seems the relationship between $J_{obs}$ and $GR$ is not clear. $J_{obs}$ ranged from 0.3 to 23.6 cm$^{-3}$ s$^{-1}$, with the mean value of 6.1 cm$^{-3}$ s$^{-1}$. $GR$ ranged from 1.1 to 8.9 nm h$^{-1}$, with the mean value of 4.2 nm h$^{-1}$. The mean $J_{obs}$ of NPF events under clean, polluted, BD and FD conditions was 5.7, 8.7, 6.6 and 6.8 cm$^{-3}$ s$^{-1}$, respectively. The corresponding mean $GR$ value was 4.2, 4.3, 4.1 and 3.7 nm h$^{-1}$, respectively, and mean $CS$ was 0.006, 0.020, 0.005 and 0.005 s$^{-1}$. $J_{obs}$ under polluted conditions was significantly higher than those under other conditions, suggesting there were abundant condensing vapours participating nucleation and overcoming the competition with the pre-existing particles (as indicated by high $CS$ value).

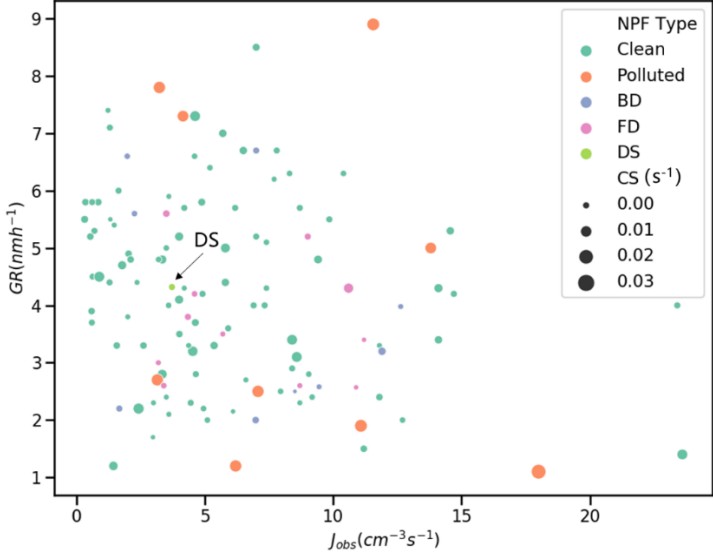

Fig.3 Scatter plot of formation rate ($J_{obs}$), grow rate ($GR$) and condensation sink ($CS$) as categorized by different NPF event types, including the cases occurring under clean, polluted conditions and influenced by blowing dust (BD), floating dust (FD) and dust storm (DS).

The statistical results of $J_{obs}$, $GR$ and $CS$ of different NPF types were given in Fig. 4. Due to the limited samples of each dust-related NPF events, the statistical analysis was conducted for all dust-related NPF events. The NPF event studies have been conducted extensively since 1990 worldwide and 2000 over China (e.g., Kerminen et al., 2018; Chu et al., 2019). Based on the one-year study of NPF events at Peking University (PKU) site in Beijing in 2004, it has been reported that $J_{obs}$ at 3 nm ranged from 3.3 to 81.4 cm$^{-3}$s$^{-1}$, and $GR$ ranged from 0.1 to 11.2 nm h$^{-1}$, respectively (Wu et al., 2007). Wang et al. (2013) has also reported $J_{obs}$ at PKU site ranged from 2.2 to 34.5 cm$^{-3}$s$^{-1}$, and $GR$ ranged from 2.5 to 15.3 nm h$^{-1}$ from March to November in 2008. PKU site locates 5 km to the north of CAMS site, which is a representative urban site in Beijing with long-term study of NPF events. Based on the long-term study at PKU site (2013-2019), it has been recently reported the annual average of $J_{obs}$ decreased from 12 cm$^{-3}$s$^{-1}$ in 2013 to 3 cm$^{-3}$s$^{-1}$ in 2017, whereas increased to 5.0 cm$^{-3}$s$^{-1}$ in 2019, and $GR$ values kept stable around 2.0–4.0 nm h$^{-1}$ during these years (Shang et al., 2022). The mean values of $J_{obs}$ and $GR$ in this study was 6.1 cm$^{-3}$s$^{-1}$ and 4.2 nm h$^{-1}$, which was comparable with the values reported by Shang et al., (2022). However, due

to the limited cases of BD, FD and DS influenced NPF events in this work, a confident comparison result among different dust related NPF events could not be derived.

    The median $CS$ of dust-related NPF, clean and polluted NPF events was 0.005 s$^{-1}$, 0.006 s$^{-1}$ and 0.020 s$^{-1}$ (Fig. 4c), respectively. It should be noted that the $CS$ calculation did not consider coarse mode particles as dust particles had faded when NPF started. Furthermore, the influence of the dust particles on $CS$ is minor based on the calculation in the following

discussion. Although the precursors from anthropogenic emissions have been proved to participating the particle nucleation and growth processes, it is difficult to quantify its contribution, especially in megacities like Beijing (Kulmala et al., 2022). The complex primary emissions, for example, traffic emissions with plentiful nanoparticles, can mix with the freshly nucleated particles, making it difficult to resolve the particles from primary emissions and secondary formation. However, based some long-term studies of NPF event, it has reported that the decrease of the precursors due to the emission control

strategies in China has caused formation rate reduction from 2013 to 2017 both in Beijing (Shang et al., 2022) and rural site in Yangtze River Delta region (Shen et al., 2022).

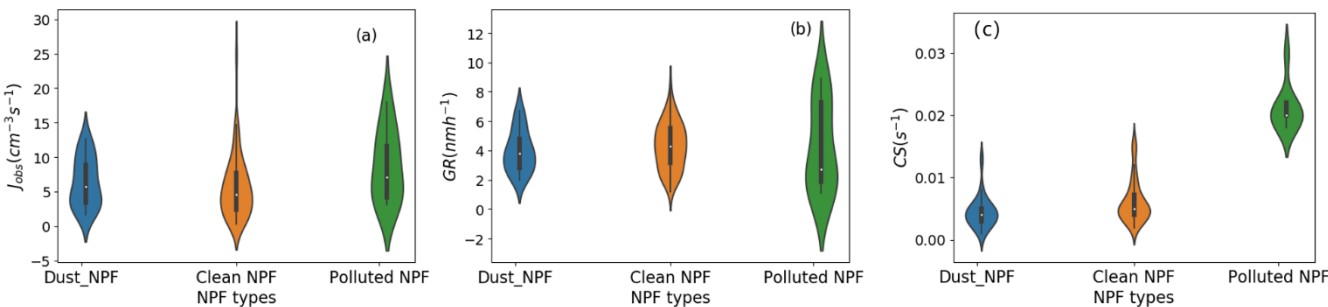

Fig. 4 The violin plot of formation rate ($J_{obs}$) (a), growth rate ($GR$) (b), and condensation sink ($CS$) of dust-related NPF (Dust_NPF) and other NPF events under clean (Clean NPF) and polluted conditions (Polluted NPF). The marker represents

the median value; a box indicating the interquartile range, and the shaded area represents the distribution probability.

    The mean diurnal concentration of particles in the size range of 3–25 nm ($N_{3-25}$) and $D_{p,g}$ on NPF events occurred under clean, polluted conditions and dust-related NPF events as influenced by BD, FD and DS (March 16, 2021), respectively, was given in Fig. 5. $N_{3-25}$ showed similar diurnal pattern, which peaked around noon time governed by NPF event. The lower $N_{3-25}$ was found on DS-related NPF event, with also smaller $D_{p,g}$ at the initial growth stage below 20 nm, indicating less precursors

participating nucleation and growth processes. It should be also addressed that only one DS-related NPF event occurred (March 16, 2021) in this study, which could not represent the general characteristics of NPF events influenced by dust storm. $D_{p,g}$ on the dust-related NPF events (including BD, FD and DS type) in Fig. 5b was generally lower than that on the clean and polluted NPF events with nucleated particles below 20 nm. A quick growth of nucleated particles at round 19-20:00 LT was probably associated with the wind direction change as given in the SM (Fig. S9). The previous studies in Beijing have revealed that the

southerly air masses containing plentiful anthropogenic precursors, facilitating the nucleation and growth processes of NPF

events (Wang et al., 2013; Shen et al., 2018). It was also found polluted-NPF events usually started later, at around 10:00 LT, with $N_{3-25}$ quickly peaked at around 12:00 LT, indicating a shorter nucleation process with larger formation rate as shown above.

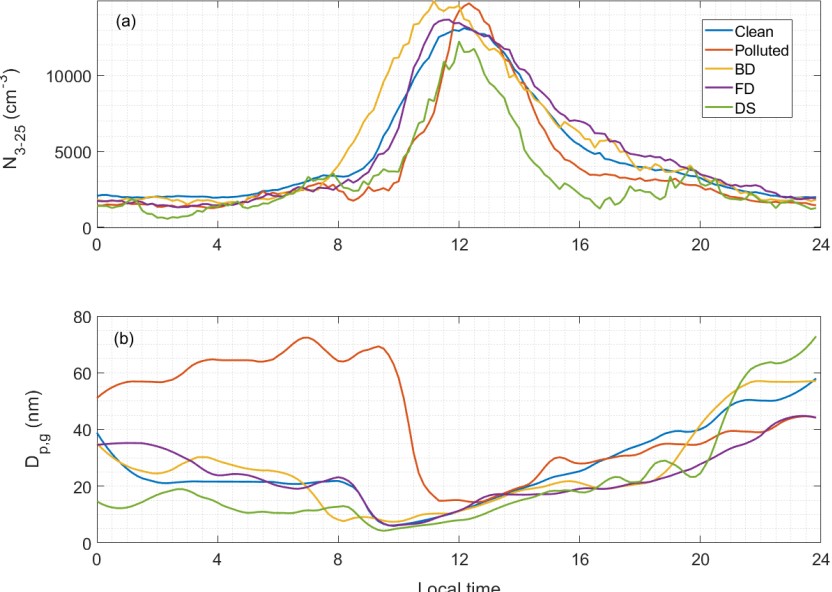

Fig. 5 the mean diurnal variation of number concentration of particles in the size range of 3-25 nm ($N_{3-25}$) (a) and geometric mean diameter ($D_{p,g}$) (b) of NPF events occurred under clean, polluted conditions and influenced by blowing dust (BD), floating dust (FD) and dust storm (DS), respectively.

## 3.2 Case study of a dust-related NPF event

### 3.2.1 A severe dust storm case

A severe dust storm that originated from Mongolia and swept through northern China on March 15, 2021, resulted in an extremely high particle mass concentration and low visibility. Four typical air quality monitoring sites, including Guanyuan (GY), Wanshou Temple (WST), Dongsi (DS), and Chaoyang (CY), in urban Beijing were selected to help understand the evolution of $PM_{2.5}$ and $PM_{10}$ mass concentrations during the dust storm (Fig. 6). The maximum of hourly mean $PM_{2.5}$ and $PM_{10}$ mass concentrations exceeded 600 and 8000 μg/m$^3$, respectively, at these selected sites which demonstrates the magnitude of this dust storm. The reported $PM_{10}$ mass concentration was the highest among the recent 20-year data in Beijing released by the CNEMC, as well as in Northern China. The $PM_{10}$ mass concentration was a magnitude higher than the value of $PM_{2.5}$, during the dust-dominated period of 8:00–18:00 local time (LT), indicating the major contribution of dust particles with size above 2.5 μm. From the PNSD plot in Fig. 7, it was found that particles above 400 nm started to increase on March 15 at 8:00 LT, indicating the arrival of dust particles. In the following two hours, the volume size distribution showed that concentration peaked in the size range of 8–10 μm. Based on the lognormal fitted parameters of the volume distribution, volume median mobility diameter ($D_{p,vol}$) was 8–10 μm during the initial stage (10:00–12:00 LT) of the dust storm and decreased to

approximately 4–6 μm from 12:00 to 24:00 LT on March 15. It was comparable with previously reported values for other dust storms across the globe, ranging from 3.0 to 6.5 μm (Reid et al., 2008; Maring, 2003; Peters, 2006).

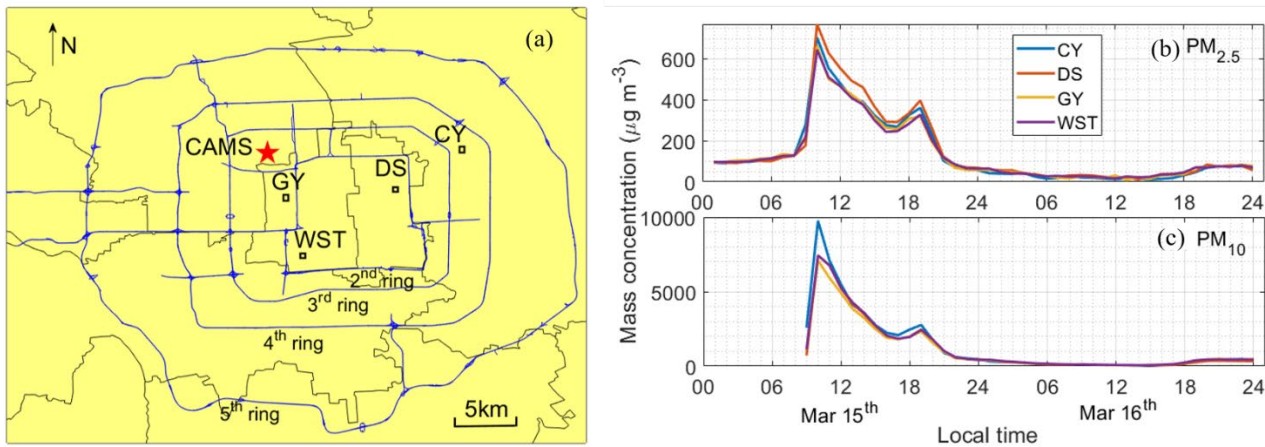

Fig. 6 The location map of CAMS site and four air quality monitoring sites, including Guanyuan (GY), Wanshou Temple (WST), Dongsi (DS), and Chaoyang (CY), in urban Beijing (a), as well as $PM_{2.5}$ and $PM_{10}$ mass concentration at each location (b).

The dust storm weakened in the late afternoon of March 15, and NPF event occurred on the morning of March 16. During the polluted conditions before the dust storm (March 14), fine particles (diameter < 1 μm) dominated the particle number and mass concentration. However, coarse-mode particles contributed the most to the particle mass, volume, and surface concentration. In contrast to the NPF events occurring during the dust storm at Mt. Heng in South China (Nie et al., 2014), the NPF event on March 16 in this study occurred when dust particles vanished and there was a reduced *CS*. However, the NPF event was interrupted in the afternoon (~16:00 LT), which was influenced by the backflow of dust, as indicated by the elevated volume concentration as shown in Fig. 7b. The concentrations of $SO_2$ and $O_3$ increased in the morning of March 16, as shown in Fig. 8, indicating enhanced precursors and atmospheric oxidation capacity that favored NPF.

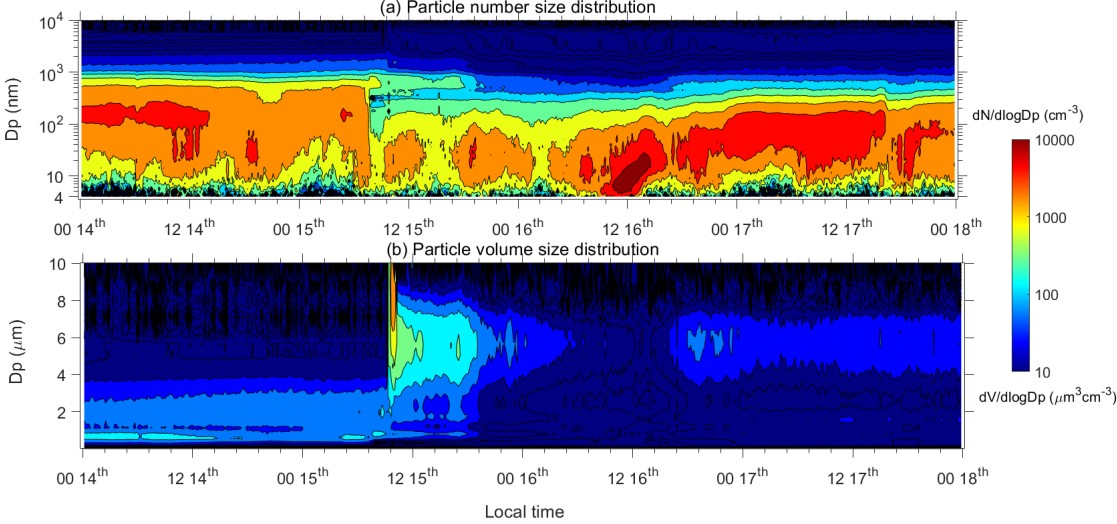

Fig. 7 Particle number (a) and volume (b) distributions on March 14–17, 2021, at CAMS station.

### 3.2.2 Secondary aerosol formation during dust storm

The variation in $NO_2$ sharply increased in the early morning (5:00–6:00 LT) on March 15, which could be attributed to the downward mixing of $NO_2$ rich air in the residual layer where $NO_2$ was trapped during the pollution episode on March 14 (Fig. 8). The wind field distribution depending on the air pressure level based on the reanalysis meteorological data was analyzed and given in Fig. S10 in SM (https://cds.climate.copernicus.eu/, latest access: June 1, 2023). It showed wind changed on 3:00 LT March 15, with the wind direction switching from upward to downward, which could bring the pollutants to the ground, resulting in the elevated $NO_2$ and *CS* as given in Fig. 8. The concentration of $SO_2$ remained stable before 6:00 LT, probably because its distribution was uniform in the boundary layer. When the dust particles arrived in Beijing at 8:00 LT on March 15, the volume mixing ratio of $NO_2$ and $SO_2$ decreased immediately due to the strong dilution by the wind. The volume mixing ratio of $NO_2$ decreased, while that of $O_3$ increased, indicating that the removal of $NO_2$ was helpful for the elevated $O_3$ concentration, as $NO_x$-titration photochemistry process could influence the production and loss of $O_3$ (Lu et al., 2010). It has been also reported that in Beijing-Tianjin-Hebei region, the decrease in $NO_x$ increased ozone and enhanced the atmospheric oxidizing capacity (Huang et al., 2020).

As shown in Fig. 8, the $PM_1$ mass concentration derived from the AMS data showed high mass loading during the pollution episode (00:00–06:00 LT on March 15), with a mean value of 83.2 μg m$^{-3}$. Secondary inorganic aerosols were dominant, with nitrate being the major contributor, accounting for approximately 46% of the total particles. During the dust storm and post-dust period, $PM_1$ sharply decreased to approximately 5.0 μg m$^{-3}$, indicating a strong fine particle scavenging process by mineral dust. In contrast to the pollution episode, organics accounted for over 50%, with a larger mass fraction of POA, approximately 40–60% of the organics during dust and post-dust storm periods. Under extremely clean conditions, secondary sulfate, nitrate, and ammonium (SNA) accounted for approximately 20%, 10%, and 15%, respectively. The chemical composition of fine particles changed from the dominant role of nitrate during the pollution episode to the largest contributor of organics during the dust storm. However, the sulfate mass fraction during dust and post-dust was higher than that during the pollution episode, indicating a higher formation process of secondary sulfate. In the previous studies in Beijing, it has been reported the NR-$PM_1$ derived from AMS could reach ~200 μg m$^{-3}$ during polluted episode and decreased to several μg m$^{-3}$ under clean conditions (Zhang et al., 2018). In this work, the $PM_1$ mass concentration ranged from approximately 5.0 μg m$^{-3}$ during dust and post dust period and to 83.2 μg m$^{-3}$ during a moderate polluted conditions before dust. During the dust and post dust period, organics was the dominant contributor to the chemical composition, which was consistent with the previous studies that organics could contributed 40-60% to $PM_1$ in Beijing (Zhang et al., 2018; 2019). However, the mass fraction of nitrate and ammonium was larger during polluted condition, which was consistent with a recent study in Beijing in January - February, 2021 (Zhang Y. et al., 2023). From 5:00 LT on March 15 to 12:00 LT on March 16, the northwesterly air mass and local wind direction did not change (Figs. S9 and S11 in SM). The $SO_2$ volume mixing ratio increased quickly when the dust storm faded (22:00 LT on

March 15), which was probably due to the weakened dilution process as the wind speed decreased from 5.8 m s$^{-1}$ during dust

episode to below 3.0 m s$^{-1}$ at 20:00 LT on March 15. Another possibility was that dust particles could be a major $SO_2$ sink (Usher et al., 2003). $NO_2$ also decreased when the dust storm started; however, it did not increase significantly when the dust storm ended, which differed from the variation in $SO_2$. However, heterogeneous reactions of mineral dust with $SO_2$, $NO_2$ can not be discussed further, due to the lack of direct measurement of chemical information of coarse mode particles.

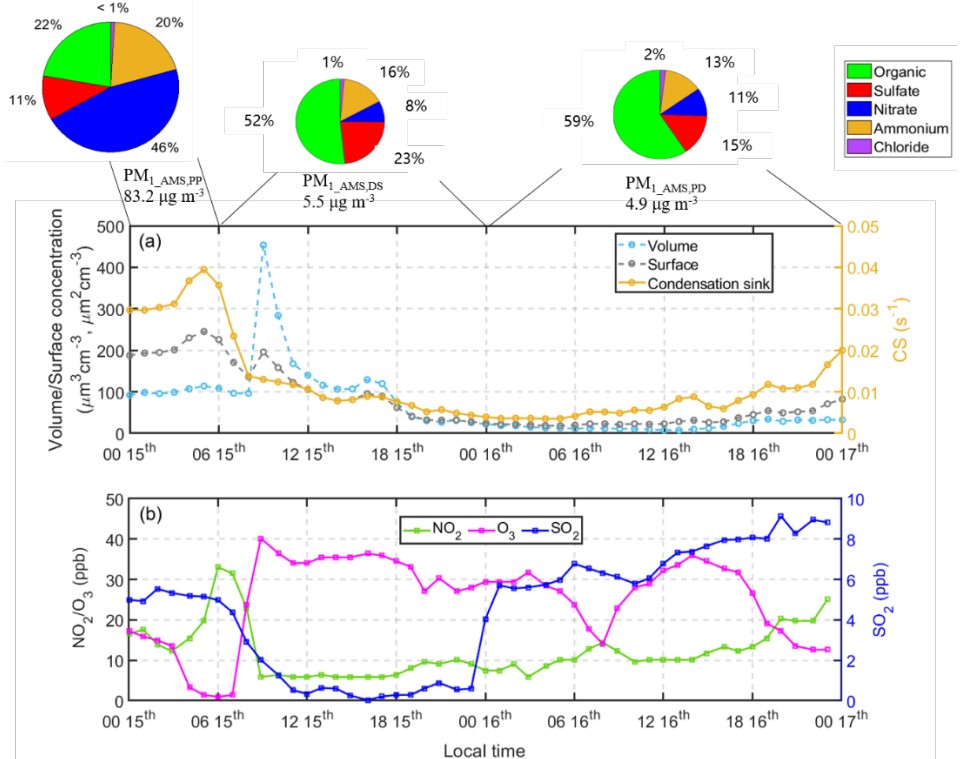

Fig. 8 Time series of $PM_{10}$ volume, surface and condensation sink from March 15–17. Pollution period (PP), dust storm (DS) and post-dust (PD) are marked in the plot and the $PM_1$ mass concentration, as well as each chemical composition (organic, sulfate, nitrate, ammonium, and chloride) mass fraction is also given for each period (a), and hourly time series of $NO_2$, $O_3$ and $SO_2$ (b).

The oxidation ratios of $SO_2$ (SOR) and $NO_2$ (NOR) are shown in Fig. 9. It has been reported that the SOR was 0.18 in clean

air conditions, whereas it was 0.27 under polluted conditions in Beijing in 2016 wintertime, indicating that $SO_2$ secondary transformation was a major pathway of sulfate production with a higher conversion efficiency under the polluted episode, whereas NOR was approximately 0.08, under both clean and polluted conditions (Wu et al., 2019). The SOR and NOR of $PM_1$ showed a clear positive relationship with the sulfate and nitrate mass concentrations, respectively, during the heavy dust period, as indicated by the volume mean diameter ($D_{p,vol}$) (Fig. 9). Although SOR and NOR were calculated based on the submicron

particles and $D_{p,vol}$ represented the dust particles, their correlations could reflect how the heterogeneous reactions were modified by the dust (Usher et al., 2003). SOR ranged from 0.2 to 0.9 during the dust storm, which was even higher than the value during the polluted episode, approximately 0.3. NOR showed a stronger positive relationship with nitrate mass

concentration. However, the NOR was relatively low, with a value of 0.01–0.1, which was much lower than the value under

polluted conditions, ranging from 0.3–0.5. This indicates that the dust particles promoted the secondary inorganic formation

of submicron particles, particularly sulfate formation. Although the uptake and oxidation of $NO_2$ and $SO_2$ by mineral dust can

be considerable, secondary aerosol formation in the submicron range is also important. As discussed above, it has been shown

that $O_3$ increases during a dust storm (Fig. 8), which indicates enhanced oxidation capability during the dust storm and favors

secondary inorganic aerosol formation by ozone oxidation. $O_3$ can also promote the heterogeneous oxidation of other trace

gases on mineral dust surfaces (Li et al., 2006; Wu et al., 2011). Although the secondary formation of sulfate and nitrate was

enhanced during the dust event, the $PM_1$ mass concentration, as well as each chemical component, decreased sharply owing

to dilution by the strong wind during the dust storm.

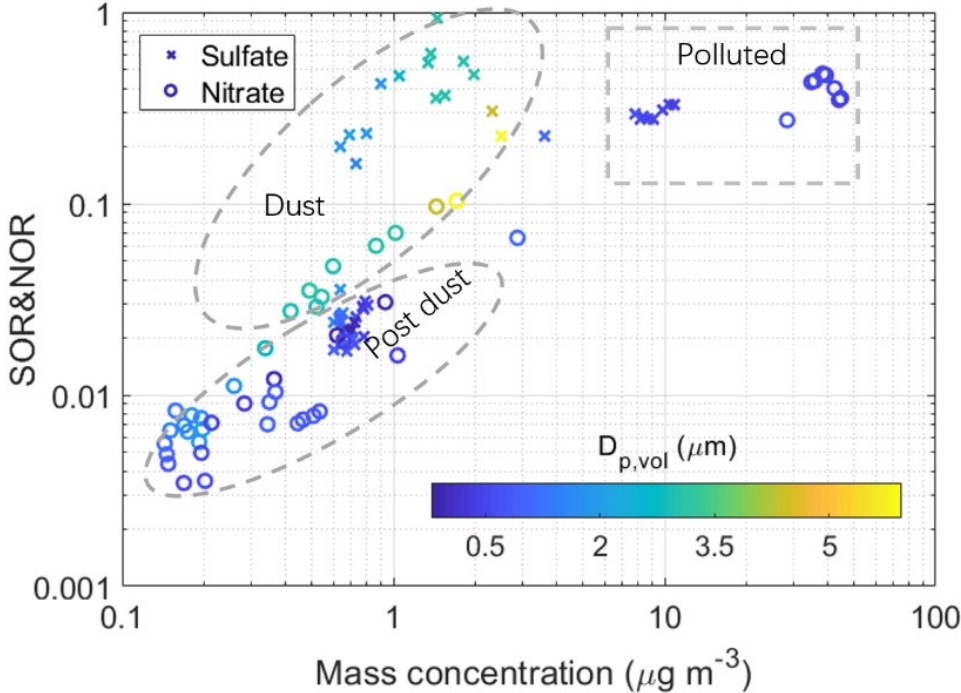

Fig. 9 Scatter plot of sulfate (cross) and nitrate (circle) mass concentrations in x-axis versus sulfur oxidation ratio (SOR)

and nitrogen oxidation ratio (NOR) in y-axis with different volume weighted diameter ($D_{p,vol}$) indicated by the color bar.

Dust, post-dust and polluted periods are marked by dashed lines, respectively.

### 3.2.3 Variations of particle hygroscopicity

Because RH was quite low during the dust storm, ranging from 10–20%, the hygroscopic growth process of the particles in

the ambient environment could be ignored. However, as discussed above, the chemical composition of $PM_1$ derived from AMS

changed significantly during the dust storm compared to the pollution episode. The calculated hygroscopic parameter, $\kappa$ values,

of 50- and 100-nm particles were significantly higher during the pollution episode than during the dust and post-dust storm

(including NPF event) periods (Fig. 10). $\kappa$ of 50 nm ranged from 0.05 to 0.17, and 0.15–0.30 for 100 nm particles, which was

consistent with the long-term study in Beijing as reported by Zhang S. et al. (2023) and Wang Y. et al. (2018). Although the

chemical composition of ultrafine particles below 100 nm can not be derived in this work, based on a recent study in Beijing,

it has been revealed that organics dominated the mass concentration of particles below 100 nm, whereas the mass fraction of
nitrate increased depending on the size (Li et al., 2023). For submicron particles in this study, the mass fraction of organic
aerosols (OA) with weak hygroscopicity accounted for approximately 20% a minor fraction of the particles during the pollution
episode, whereas it increased to more than 50% during the dust storm and post-dust storm periods (Fig. 10c). Meanwhile, the
fraction of hydrophobic POA increased during the dust storm and post-dust storm periods (Fig. 10d), as strong wind diluted
the pre-existing particles, and the anthropogenic emission contribution was less important during the severe dust storm.
Although sulfate formation was enhanced during the dust storm, as discussed above, with the larger mass fraction during the
dust and post-dust storm periods, the smaller fraction of nitrate and higher fraction of OA resulted in weaker particle
hygroscopicity during the dust and post-dust storm periods. This indicates that dust storms can modify the chemical
composition of submicron particles, which influences their hygroscopic behavior and ability to be activated as CCN. During
post-dust period, NPF event occurred on March 16, which produced a large quantity of nano-particles as the potential CCN.
The particle number concentration increased and the particle hygroscopicity decreased as compared with the polluted episode,
and the variation of CCN concentration were evaluated further in the following discussion.

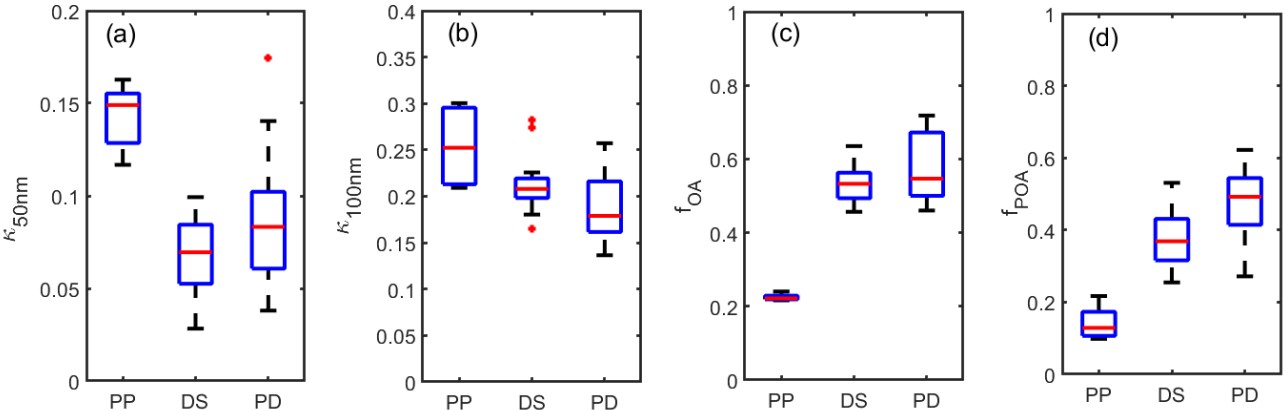

Fig. 10 Statistical boxplot of hygroscopic parameter ($\kappa$) of 50 (a) and 100 nm (b) and the mass fraction of organic aerosol
(OA), $f_{OA}$ (c) accounting for $PM_1$, and mass fraction primary OA accounting for the organics, $f_{POA}$ (d) during the pollution
(PP), dust (DS) and post-dust (PD) periods. The upper and lower boundaries of the box plots indicate the 75[th] and 25[th]
percentiles, respectively, the line within the box marks the median, and the whiskers above and below the box indicate the
90th and 10th percentiles, respectively. Data points beyond the whiskers are displayed using a red cross.

### 3.2.4 Impact on the cloud condensation nuclei by the dust storm

Assume that $\kappa_{htdma} = \kappa_{ccn}$, $D_{p,crit}$ can be calculated based on Equations (6) and (7) on March 15 and 16, with a $\kappa$ value of 100
nm dry particles at a low or moderate cloud supersaturation ($S_c$) of 0.2% and a high $S_c$ of 0.7%. The $D_{p,crit}$ ranged from 45 to
60 nm at $S_c = 0.7\%$, with a mean value of $49 \pm 2.5$ nm, $52 \pm 2.0$ nm, and $53 \pm 2.6$ nm, during the polluted, dust, and post-dust
storm periods, respectively, which were $112 \pm 5.5$ nm, $119 \pm 6.5$ nm, and $123 \pm 7.2$ nm, during the corresponding periods at $S_c$
$= 0.2\%$. $D_{p,crit}$ increased during the dust storm and post-dust storm periods, by approximately 6% and 10%, respectively,

compared with the pollution episodes. As $D_{p,crit}$ was larger than 45 nm, the number of particles larger than 45 nm ($CN_{45}$) was calculated and referred to as the potential CCN. The concentration of particles with diameters larger than $D_{p,crit}$ was calculated as the potential condensation nuclei ($CCN_{cal}$). The scatter plot of $CN_{45}$ and $CCN_{cal}$ under different conditions, indicated by the $D_{p,vol}$, as well as the activation ratio values ($R$) calculated by $CCN_{cal}$ dividing $CN_{45}$, are shown in Fig. 11. It shows $R$ values ranging from approximately 0.1 to 0.8 with $S_c = 0.2\%$, with higher values during polluted episodes (~0.8), lower values during the dust storm (0.2–0.4), and even reached 0.1–0.2 during the post-dust period. However, at high $S_c = 0.7\%$, $R$ values ranged from 0.6–1.0, and were close to 1.0, during the polluted episode, and concentrated around 0.8 during the dust episode. Results showed that the CCN activation capability was significantly influenced by the dust storm at a low $S_c$, whereas the influence was minor at a high $S_c$, depending on $D_{p,crit}$. Although an NPF event occurred during the post-dust period on March 16, $CCN_{cal}$ did not increase because of the elevated critical diameter. Although the backflow of floating dust particles did not result in clear increase of $PM_{2.5}$ and $PM_{10}$ mass concentration given in Fig. 6b, c, they induced a higher $CS$ (Fig. 8a), which was not favorable for the further growth of the nucleated particles.

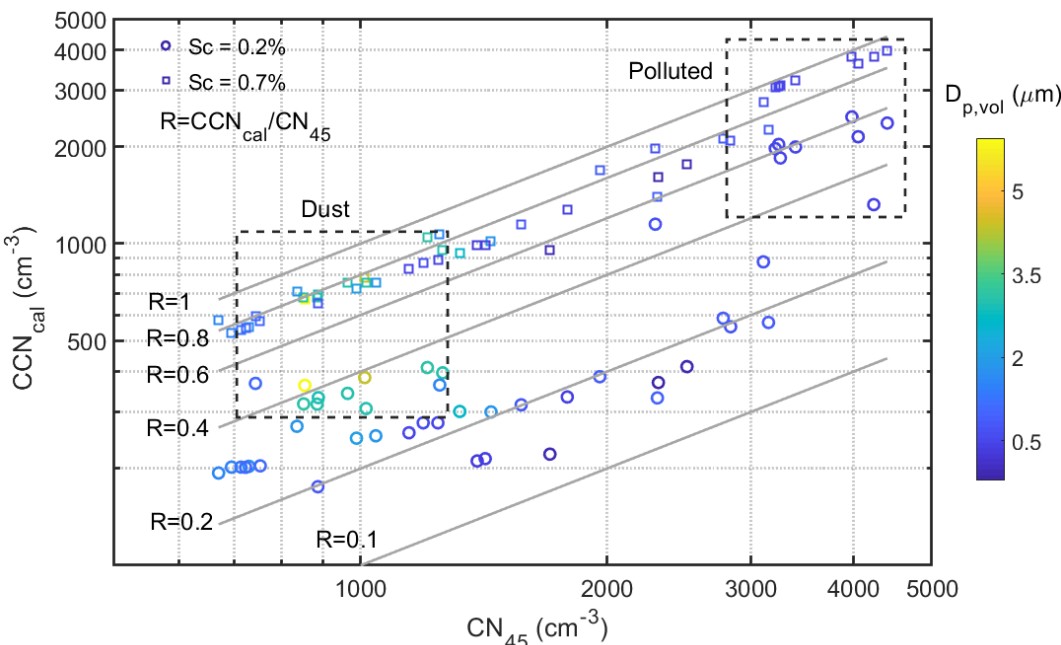

Fig. 11 Scatter plot of calculated cloud condensation nuclei ($CCN_{cal}$) under different supersaturations (Sc = 0.2% and 0.7%) and number concentration of diameter above 45 nm ($CN_{45}$), with different volume weighted diameter ($D_{p,vol}$) indicated by the colors. The solid lines representing the different ratio values between $CCN_{cal}$ and $CN_{45}$.

## 4    Conclusions

Dust days, including dust storms, blowing dust, and floating dust, have occurred frequently in the spring in recent years. We found there were approximately 80% of dust days followed by the NPF event based on the PNSD measurement in Beijing in spring from 2017 to 2021. Owing to the scavenging effect of strong winds on pre-existing aerosols, the condensation sink

remained at a quite low value of ~0.005 s$^{-1}$, which favored the occurrence of NPF events after dust events. The NPF events were classified into cases following dust (dust-related NPF) and other NPF events occurring under clean and polluted conditions. Based on the PNSD derived on NPF days, it was found that the observed formation and growth rates were approximately 50% of and 30% lower than those on other NPF events, respectively.

The most severe mineral dust storm over the past two decades in China originated in Mongolia, and swept through northern China on March 15–16, 2021, with the highest hourly mean PM$_{10}$ reaching 8 mg m$^{-3}$, as observed in urban Beijing. The processes of mineral dust and their impact were evaluated based on field measurements of aerosols and reactive gases in urban Beijing. During dust storms, the volume-weighted particle size peaks at approximately 4–6 μm or 8–10 μm. In particular, secondary sulfate aerosol formation was found to have a higher rate than that of the polluted episode before the storm. The hygroscopicity of ultrafine particles was weakened during the dust and post-dust storm periods, as compared with the pollution episode, due to the elevated mass fraction of organics, especially the primary organic aerosols. Based on the $\kappa$-Köhler theory, the critical diameter of the particles activated as CCN was calculated for different supersaturations. They were approximately 49 nm and 112 nm in the polluted episode at $S_c$ of 0.7 and 0.2%, respectively, which increased to 52 nm and 119 nm, respectively, during the dust storm, increasing by approximately 6%. A new particle formation event occurred on March 16, when the dust vanished; however, the contribution to the CCN concentration was minor. As a consequence of the uptake of precursor gases on mineral dust, the physical and chemical properties of submicron particles changed, thereby influencing their ability to act as CCNs, especially at low supersaturation. The impact of this severe dust event observed in Beijing provided valuable information for evaluating the influence of dust, particularly the underlying impact of submicron particles and their ability to be activated as CCN. However, more work is needed to quantify the contribution of anthropogenic emissions to NPF based on field experiments and modelling work in the future.

**Acknowledgments**

This research was supported by the National Natural Science Foundation of China (42075082, 41825011, 42090031), S&T Development Fund of CAMS (2020KJ001), Basic Research Fund of CAMS (2020Z002), and Innovation Team for Haze-fog Observation and Forecasts of MOST and CMA.

**Author contributions**

XS planned the study, conducted the measurement, analyzed the data, and wrote the original draft. YZ, QL, CZ, KG, JZ, SZ, WX, XH, JL, SL, JW, AY and CX contributed to the measurement, instrument maintenance, data analysis and result discussion. JS, HC and XZ reviewed and finalized the article. XS and JS contributed to fund acquisition.

**Competing interests**

The authors declare no conflict interest.

**Data availability**

All data used in the study are available from the corresponding author upon request (shenxj@cma.gov.cn).

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
