# Peer review of "Characterization of dust-related new particle formation events based on long-term measurement in North China Plain"

_Atmospheric Chemistry and Physics, 2022_

## Author Comment (AC1)

Response to reviewers

Thanks for your great efforts and valuable comments, which helps to improve our manuscript. We have addressed the reviewers' comments on a point-to-point basis as below for consideration. Referee comments are in black. Author responses are in red. All the line numbers mentioned following are refer to the revised manuscript with no changes marked.

This manuscript is built on continuous aerosol measurements conducted during 4 springs in urban Beijing. The data and research topic are certainly of high interest. While the paper is well-organized and technically relatively well written, there are a few major issues that prevent me to recommend the publication of this paper in its present form. My main concerns in this regard are summarized below.

My first major comment is about the imbalance between the title of the paper and its contents. The title gives an impression that this paper is solely about association between new particle formation (NPF) and dust, but most of the results (sections 3.3-3.5 and a big fraction of section 3.2) have little/nothing to do with NPF. The same concerns the abstract.

Reply: Thanks for your helpful comments. In section 3.1-3.2, we generally described the dust days and NPF occurrence probability in spring from 2017 to 2021. The classification of two types of NPF events was conducted depending on whether the dust event occurred before NPF, to evaluate the influence of dust process on nucleation and growth process. In the section 3.3-3.5, we discussed almost the most severe dust storm case in recent twenty years and the following NPF case. For this case, we have relatively comprehensive measurements of particle size, chemical composition, hygroscopicity, which can help to reveal the variations of physical and chemical properties of nucleated particles, and even the ability to be activated as CCN. However, as reviewer mentioned, we should highlight the relationship with NPF in each section, to make the discussion more focused on how the dust particles modify the atmospheric conditions when NPF events occur.

One of the main results brought up by the authors is the contribution of anthropogenic emissions to particle formation and growth rates. Estimating such contributions is very difficult overall, and totally impossible when having no real-time measurements of low-volatile precursors causing NPF and growth. Simply comparing days with and without presence of dust cannot address this issue, so in this regard the conclusions made in this paper are not scientifically sound.

Reply: The influence of anthropogenic emissions on NPF event is difficult to be evaluated or quantified, especially in urban areas (Kulmala et al., 2022). In this work, we analyzed the long-term measurement of PNSD in spring in Beijing from 2017 to 2021, to characterize the NPF event influenced by dust process or not. However, as the reviewer mentioned, there is no sufficient percussor measurement, especially the lowvolatile organic vapor. Thus, we revised the discussion and abstract, to address the influence of dust event on NPF, instead of accessing the contribution from anthropogenic emissions. For example, in the abstract, the description about "contribution of anthropogenic emissions" has been revised to "By comparing the two types of NPF events, the observed formation ($J_3$) and growth rate ($GR$) of dust-related NPF events were approximately 50% and 30% lower than the values of normal NPF days, respectively, due to the extremely low condensation sink (~0.005 s$^{-1}$) caused by the strong wind during the dust process. The difference of NPF parameters got smaller when nucleated particles grew into the sizes above 10 nm, as the anthropogenic emissions accumulated fast during the few hours when dust ended and favored the growth process in the later stage." We also revised the discussions and conclusions.

As noted by the authors, condensation sink (CS) is an essential parameter determining whether NPF is possible in various conditions. It is a pity that the authors did not calculate CS for course mode particle, as it would have been possible from APS measurements. The total CS was very likely dominated by dust particles at least during the heaviest dust storms, and this might be one explanation why NPF was observed after the storms rather than during them.

Reply: The authors totally agree with the reviewer's comment that the coarse mode particles are major sink for precursor gases, which can result in high condensation sink ($CS$). Thus, NPF events do not occur during dust period. When dust particles fade, the $CS$ is quite low (~0.005 s$^{-1}$) due to strong dilution by wind, which favored for the NPF event occurrence. That means when we talk about NPF, the dust period has ended and the concentration of pre-existing particles is quite low, and thus the contribution of coarse mode particle to $CS$ can be ignored. Unfortunately, the APS measurement was not continuous and only available on March 2021. We discussed a typical and most severe dust case during 2017-2021. For this case study, the APS data was available and the influence on $CS$ is discussed. It was found $CS$ during dust storm was <0.02 s$^{-1}$, whereas it was much lower than the value during air polluted conditions (~0.04 s$^{-1}$) before dust and decreased to be lower than 0.01 s$^{-1}$ when NPF occurred, as shown in Fig. 8 in the manuscript. In this study, we focused on how the dust event modified the atmospheric conditions when NPF occurred.

There are potentially valuable data on non-NPF-related chemistry associated with dust/non-dust in sections 3.3-3.5. Unfortunately, the current discussion on these data is rather qualitative, relying mainly on finding reported by earlier literature, and providing little new scientific insight. For example, the statement on lines 304-305 is self-evident. The authors did not explain how they combined direct hygroscopicity measurements to the hygroscopicity estimated from measured aerosol chemical composition.

Reply: In this study, we provided valuable information about how the dust processes modify the atmospheric conditions which facilitate the NPF event, which has been rarely discussed in the open literature. However, due to the limited measurement data,

some discussions are not thoroughly to give the scientific insight. We have supplemented discussions in the manuscript, to make the scientific conclusions more robust. For example, a profound discussion about how different dust types influenced NPF events was given and addressed the strength of dust processes determined the condensation sink before NPF event, which was a key parameter in determining the formation and growth rate of NPF.

The sentence of original line 304-305 has been removed. In this study, the hygroscopicity parameter ($\kappa$) is derived from H-TDMA directly, which was not estimated from chemical composition data from AMS. The variation of $\kappa$ was consistent with the chemical component.

Minor issues:

lines 43-44: reduced compared to what?

Reply: the sentence has been corrected to "Model simulations were performed with and without dust, and the results predicted that total particle concentration and CCN were reduced by approximately 20% and 10%, respectively, as influenced by the dust pollution plume in East Asia (Manktelow et al., 2010)."

lines 54-55: 45% of aerosols. By what measure? AOD?

Reply: This sentence has been corrected to "However, based on the optical parameters, including particle linear depolarization ratio, volume linear depolarization ratio and lidar ratio derived from a Raman lidar, there were approximately 45% of aerosols below 1.8 km above the ground contributed by polluted dust (the mixture of anthropogenic aerosols and dust) in Northern China (Wang et al., 2021)."

line 305: this should be Fig. 10, not Fig. 8.

Reply: The hygroscopic parameters are given in Fig. 10, and the chemical composition are given in Fig. 8.

It is somewhat unclear what is the difference between positive and negative anomalies in Figure 3. Also, it not well explained what is subtracted from what in this figure.

Reply: We have supplemented the details of Fig. 3 in the text, to make it clear that how the anomalies are calculated. The anomaly plots were obtained by means of the PNSD of NPF occurring on non-dust days subtracting the mean PNSD of dust-related NPF in each spring from 2017 to 2021, and was shown in Fig. 3. The positive anomaly indicated how much the particle number concentration in each corresponding size bin on non-dust NPF days was higher than that on dust-related NPF days, whereas the negative anomaly indicated that PNSD was lower on non-dust NPF days.

Finally, it seems to me that not all references cited in the text can be found in the reference list.

Reply: The authors check through the manuscript and supplemented the missing references in the list.

Dupart, Y., King, S. M., Nekat, B., Nowak, A., Wiedensohler, A., Herrmann, H., David, G., Thomas, B., Miffre, A., Rairoux, P., D'Anna, B. and George, C.: Mineral dust photochemistry induces nucleation events in the presence of $SO_2$, Proc Natl Acad Sci USA, 109(51): 20842-20847, DOI: 10.1073/pnas.1212297109, 2012.

Hussein, T., Hameri, K., Aalto, P., Paatero, P. and Kulmala, M.: Modal structure and spatial temporal variations of urban and suburban aerosols in Helsinki Finland, Atmospheric Environment, 39: 1655–1668, DOI: 10.1016/j.atmosenv.2004.11.031, 2005.

Maring, H., Savoie, D. L., Izaguirre, M. A., Custals, L. and Reid, J. S.: Mineral dust aerosol size distribution change during atmospheric transport, J. Geophys. Res., 108(D19), 8592, doi:10.1029/2002JD002536, 2003.

Nie, W., Ding, A., Wang, T., Kerminen, V. M., George, C., Xue, L., Wang, W., Zhang, Q., Petaja, T., Qi, X., Gao, X., Wang, X., Yang, X., Fu, C. and Kulmala, M.: Polluted dust promotes new particle formation and growth, Sci Rep, 4: 6634, DOI: 10.1038/srep06634, 2014.

Peters, T. M.: Use of the aerodynamic particle sizer to measure ambient $PM_{10–2.5}$: The coarse fraction of $PM_{10}$, J. Air Waste Manage. Assoc., 56, 411–416, 2006.

---

## Author Comment (AC2)

Response to reviewers

Thanks for your great efforts and valuable comments, which helps to improve our manuscript. We have addressed the reviewers' comments on a point-to-point basis as below for consideration. Referee comments are in black. Author responses are in red. All the line numbers mentioned following are refer to the revised manuscript with no changes marked.

- **RC2**: 'Comment on acp-2022-837', Anonymous Referee #2, 06 Feb 2023

This manuscript entitled 'Characterization of dust-related new particle formation events based on long-term measurement in North China Plain', describes the observation of new particle formation events following dust episodes observed in Beijing between 2017 and 2021, with a focus on one severe episode on March 15-16, 2021. Although the topic of the paper and the potential results regarding the interaction of dust with available precursors resulting in NPF could be an important addition to literature, the authors' claims that are not supported by scientific evidence prevent me from recommending this paper for publication in ACP.

General comments:

1. The authors evaluate the contribution of anthropogenic emissions to the nucleation processes.
   The authors calculated the anthropogenic emissions as the difference in the size distribution, formation rates and growth rates during other NPF days and NPF following dust days. This approach is not justified, as it ignores available sinks, meteorology and atmospheric reactions.

   Reply: The authors agree with the reviewer's comment that quantifying the contribution of anthropogenic emissions to NPF events based on the comparison of dust related NPF and non-dust NPF days was not robust and justified, due to the lack of direct measurement of precursors and theoretical calculations. However, clear differences between dust related and non-dust related NPF events were found based on the long-term PNSD measurement, providing valuable information about how the dust storm process modified the atmosphere conditions when NPF event occurred. According to the reviewer's comments, we removed the conclusions about the quantifying of the influence of anthropogenic emissions on NPF and focused on how the dust process affecting NPF and highlight the intrinsic reasons.

2. Sections 2.3-2.5 discuss a case study in March 15-16, 2021. While the results are interesting, the conclusion cannot be extrapolated to other events as the case study described is one of a kind in the four years, as shown in Table 2. A comparison between NPF events following different types of dust episodes events could be helpful. In particular, a comparison between the episodes on March 15 and 28, with

including interpretation of NPF and its related properties could give insight on the effect of different types of dust.

Reply: Thanks for your comments, we added the discussion of NPF events followed different types of dust days. We found the dust related NPF events are highly dependent on the strength and the end time of dust process. NPF event usually occurred around 8:00-12:00 when solar radiation increased, so if the dust processes prevailed during this time, it will prevent NPF occurrence. For some cases of blowing and floating dust days overlaid by the anthropogenic aerosols, for example, March 18, 2019, April 5, 2019 and May 15, 2019, the $CS$ remains high during the dust (above 0.02 s$^{-1}$), which prevented NPF event. Under some cases (April 15-16, 2021, Fig. S1), the whole dust process was identified as blowing and floating dust episode depending on the strength how air masses containing dust particles influenced Beijing. NPF event can be observed when the whole dust process finished.

Blowing dust and floating dust was observed on May 6-7 and May 8, respectively, and both followed by the occurrence of NPF event, as shown in Fig. S2. It was also found the condensation sink ($CS$) before NPF started was approximately 0.0025-0.003 s$^{-1}$ for both NPF events, indicating the extremely weak scavenging process by the pre-existing particles. The $CS$ level were also similar during the NPF event indicating the concentration level of precursors and condensing vapor participating nucleation and growth were comparable for these two cases. The values of formation rate ($J_3$) and growth rate ($GR$) of NPF event on May 7$^{th}$ and 8$^{th}$ were quite close, with $J_3$ of 4.5 and 4.8 cm$^{-3}$s$^{-1}$, $GR$ of 2.3 and 2.5 nm h$^{-1}$, respectively.

The discussions are supplemented in the manuscript and the figures are given in the supplementary materials.

[Figure]

Fig. S1, NPF event occurred following blowing dust (April 15) and floating dust (April 16) in 2021

[Figure]

Fig. S2, NPF event occurred following blowing dust (May 6 and 7) and floating dust (May 8) in 2021. The particle number (a) and volume (b) size distribution are given and *CS* is given by pink line.

[Figure]

Fig. S3 The NPF event occurred after dust storm on March 15 (a) and blowing dust on March 28, 2021 (b), respectively. The pink line in panel indicates the condensation sink.

3. The authors report that an NPF event occurs after approximately 80% of dust episode, what was the limiting factor for the remaining 20%? Was the airmass direction different, the wind speed lower etc? Identifying the causes that inhibited the NPF during those 20% could improve the quality of the results.

Reply: Thanks for the constructive suggestion. We checked through all the dust days with NPF and without NPF days. It was found the air masses changed from northwest with the dust sources to other directions with anthropogenic emissions dominated. As given in Fig. S4-S5, floating dust occurred on May 15 2019 and April 1 2021, without

NPF event when dust diminished. It showed the air mass originated from northwest during dust period on 00 UTC and switched to southwest since 06 UTC on May 15 and southeast since 06 UTC on April 1, respectively. The change of air masses resulted in the polluted dust case, which is the mixture of anthropogenic aerosols and dust and NPF event was prohibited due to the high level of condensation sink. Even the nucleation process was observed on April 1, the growth process was interrupted by the elevated background aerosol concentration, indicated by the increasing *CS*.

The 72 hours back trajectories arriving at CAMS station at four times a day (00, 06, 12, 18 UTC) on May 15 2019 and April 1 2021, respectively, with the terminal height of 500 m above ground level, derived by the TrajStat software, combined with HYSPLIT 4 model (Draxler and Hess, 1998; Wang et al., 2009). Trajstat, is a geographic information system-based software, which can view, query, and cluster the air mass trajectories and also conduct the potential source contribution analysis (Wang et al., 2009).

The above information has been added in the manuscript and supplementary materials.

[Figure]

Fig. S4. The 72 hours back trajectories arriving at CAMS station at four times a day (00, 06, 12, 18 UTC) on May 15th 2019 (black lines) and April 1st 2021 (blue lines) with the terminal height of 500 m above ground level, derived by the TrajStat software, combined with HYSPLIT 4 model

[Figure]

Fig. S5. The cases of dust episode (May 15, 2019 and April 1, 2021) without typical NPF followed.

Draxler, R.R., Hess, G.D., 1998. An overview of the HYSPLIT_4 modeling system for trajectories, dispersion, and deposition. Aust. Meteorol. Mag. 47 (4), 295–308.

Wang, Y.Q., Zhang, X.Y., Draxler, R.R., 2009. TrajStat: GIS-based software that uses various trajectory statistical analysis methods to identify potential sources from long-term air pollution measurement data. Environ. Model. Software 24 (8), 938–939. https://doi.org/10.1016/j.envsoft.2009.01.004.

4. In section 3.3, the authors speak about primary and secondary organics, how did the authors derive this separation? Were there source apportionment techniques involved? Or was it based on previous literature? More details on the technique and results are needed.

Reply: The primary and secondary organic aerosols are identified based on the AMS data, by applying PMF method. The details are supplemented in the section 2.2 instrumentation.

Positive matrix factorization (PMF) (Ulbrich et al., 2009) and a multilinear engine (ME-2) (Canonaco et al., 2013) modelling of high time resolution organic mass spectrometric data from HR-ToF-AMS have also been used to resolve organics into primary organic aerosols (POA) and oxygenated organic aerosols (OOA), which correspond to different sources and processes (Zhang et al., 2022).

Canonaco, F., Crippa, M., Slowik, J. G., Baltensperger, U. and Prévôt, A. S. H.: SoFi, an IGOR-based interface for the efficient use of the generalized multilinear engine (ME-2) for the source apportionment: ME-2 application to aerosol mass spectrometer data, Atmos. Meas. Tech., 6(12): 3649-3661, DOI: 10.5194/amt-6-3649-2013, 2013.

Ulbrich, I. M., Canagaratna, M. R., Zhang, Q., Worsnop, D. R. and Jimenez, J. L.: Interpretation of organic components from Positive Matrix Factorization of aerosol mass spectrometric data, Atmos. Chem. Phys., 9(9): 2891-2918, DOI: 10.5194/acp-9-2891-2009, 2009.

Zhang, Y., Zhang, X., Zhong, J., Sun, J., Shen, X., Zhang, Z., Xu, W., Wang, Y., Liang, L., Liu, Y., Hu, X., He, M., Pang, Y., Zhao, H., Ren, S. and Shi, Z.: On the fossil and non-fossil fuel sources of carbonaceous aerosol with radiocarbon and AMS-PMF methods during winter hazy days in a rural area of North China plain, Environ Res, 208: 112672, DOI: 10.1016/j.envres.2021.112672, 2022.

5. I suggest that the authors revise some of the interpretations in the paper which are either not consistent or not based on evidence from the observations. Examples:

Reply: we agree with the reviewer's comments that some conclusions in the manuscript were nor robust, and we have revised them based on the measurement data directly. Some sentences are not helpful to discussion, which are removed from the manuscript.

- Line 241: "When the dust particles arrived in Beijing at 8:00 LT on March 15, the volume mixing ratio of $NO_2$ and $SO_2$ decreased immediately, which was also influenced by the enhanced aerosol surface uptake process owing to elevated particle surface concentration" vs Line 247: "However, the concentrations of $NO_2$ and $SO_2$ were low during dust storms, indicating that anthropogenic emissions had less influence."

Reply: original L241: it has been revised to "When the dust particles arrived in Beijing at 8:00 LT on March 15, the volume mixing ratio of $NO_2$ and $SO_2$ decreased immediately due to the strong dilution by wind".

original L247: this sentence was removed.

- Line 284: During the dust storm period, transitional metal ions such as Fe and Mn can act as catalysts that favor sulfate formation via $SO_2$ oxidation (Usher et al 2003).

Reply: This sentence was removed.

- Line 262: The decrease in $NO_2$ and nitrate suggests that they probably shifted from fine to coarse particles during dust storms (Wang et al., 2013).

Reply: As we do not have direct chemical composition measurement, this sentence is removed.

- Line 269: This indicates different sinks for $SO_2$ and $NO_2$ during the dust storms. As discussed above, the uptake by mineral dust was a major sink for $SO_2$, whereas the uptake of $NO_2$ was minor, as the concentration remained low when the dust storm ended.

Reply: This paragraph has been revised to "The $SO_2$ volume mixing ratio increased quickly when the dust storm faded (22:00 LT on March 15), which was probably due to the weakened dilution process as the wind speed decreased from 5.8 m/s during dust episode to below 3.0 m/s at 20:00 LT on March 15. Another possibility was indicating that dust particles could be a major $SO_2$ sink (Usher et al., 2003). $NO_2$ also decreased when the dust storm started; however, it did not increase significantly when the dust storm ended, which differed from the variation in $SO_2$. However, heterogeneous reactions of mineral dust with $SO_2$, $NO_2$ can not be discussed further, due to the lack of direct measurement of chemical information of coarse mode particles."

6. The methods section lacks detailed explanation of most of the instrumentation and quality control.

- Were the aerosol measuring instruments cross checked for over-lapping sizes? Citations are missing.

Reply: The details of instrumentation, experimental setup and data quality control, as well as the citations, are supplemented in the text.

As the reviewer mentioned, there is an overlap size range of particle number size distributions (PNSDs) derived from TSMPS and APS, which is 500-850 nm. The resulting distributions of APS system were converted from aerodynamic to mobility diameters with assumed particle density of 2.5 g/cm$^3$ during the dust, as combined with TSMPS data. The PNSDs and calculated volume size distribution derived from TSMPS and APS, respectively, during March 15-17, 2021 was given in Fig. S6. The volume concentration was calculated based on PNSD, with the assumption of spherical shape. It showed the submicron particles ($PM_1$) was the dominant contributor to the particle number concentration (a), whereas the contribution to the volume can be ignored (b). It was found in the overlap size range between TSMPS and APS data (500-850 nm), the bias became smaller when particle size increased. At the last size bin of 850 nm, the number concentration derived from TSMPS was 50% higher than that from APS. The information of the overlap size range was supplemented in the text and the figure was given in the supplementary materials.

[Figure]

Fig. S6. The number size distribution (a) and volume size distribution (b) derived from TSMPS and APS.

The description of H-TDMA measurement and data inversion has been also added. The H-TDMA system is comprised of two DMAs, a CPC (Model 3772, TSI Inc., USA) and a humidifier system between the two DMAs. The first DMA selects the quasi-monodisperse particles at a diameter ($D_{p,dry}$ = 50, 100 nm) under the dry state with 30% RH (Maβling et al., 2003). Then, the size-selected particles pass through a humidity conditioner, which can be adjusted to the setting RH of 90%. The probability distribution function (PDF) of hygroscopic growth factor (HGF), HGF-PDF is inverted by the TDMAinv method developed by Gysel et al. (2009).

Maβling, A., Wiedensohler, A., Busch, B., Neusüß, C., Quinn, P., Bates, T., Covert, D.: Hygroscopic properties of different aerosol types over the Atlantic and Indian Oceans. Atmos. Chem. Phys. 3, 1377–1397, 2003.

Gysel, M., McFiggans, G.B., Coe, H.: Inversion of tandem differential mobility analyser (TDMA) measurements. J. Aero. Sci. 134–151, 2009.

- What are the detection limits and the time resolutions of the trace gas measuring instruments?

Reply: The TE 49C has a lower detection limit of 1 ppb and a precision of 1 ppb. The 42 CTL has a lower detection limit of 50 ppt and a precision of 0.4 ppb. The 43 CTL has a lower detection limit of 0.10 ppb and a precision of 1 ppb (Lin et al., 2009). Measurement signals of trace gases were recorded as 1 min averages (Lin et al., 2011), however, the hourly average data were used for discussion, in order to match with the PM mass concentration data. The above information and references have been added in the manuscript.

Lin, W., Xu, X., Ge, B. and Zhang, X.: Characteristics of gaseous pollutants at Gucheng, a rural site southwest of Beijing, Journal of Geophysical Research, 114, DOI: 10.1029/2008jd010339, 2009.

Lin, W., Xu, X., Ge, B. and Liu, X.: Gaseous pollutants in Beijing urban area during the heating period 2007–2008: variability, sources, meteorological, and chemical impacts, Atmospheric Chemistry and Physics, 11(15): 8157-8170, DOI: 10.5194/acp-11-8157-2011, 2011.

- The description of the HR-AMS-ToF is not sufficient, and citations are missing. For example what was the collection efficiency?

Reply: The description of HR-ToF-AMS with the citations has been supplemented in the manuscript. The chemical composition of non-refractory $PM_1$, including organic components, sulfate, nitrate, ammonium, and chloride, was derived using HR-ToF-AMS with a 5-min resolution (Drewnick et al., 2005). The calibrations of ionization efficiency were performed, using size-selected (300 nm) ammonium nitrate particles before and after the experiment. Default relative IE values were used for organics (1.4), nitrate (1.1), sulfate (1.2), ammonium (4.0), and chloride (1.3). The HR-ToF-AMS collection efficiency (CE) accounts for the incomplete detection of aerosol species owing to particle bounce at the vaporizer, and/or the partial transmission of particles by the lens (Canagaratna et al., 2007). In this study, a composition-dependent CE correction was used, following the methodology described by Middlebrook et al. (2012).

Canagaratna, M. R., Jayne, J. T., Jimenez, J. L., Allan, J. D., Alfarra, M. R., Zhang, Q., Onasch, T. B., Drewnick, F., Coe, H., Middlebrook, A., Delia, A., Williams, L. R., Trimborn, A. M., Northway, M. J., DeCarlo, P. F., Kolb, C. E., Davidovits, P., and Worsnop, D. R.: Chemical and microphysical characterization of ambient aerosols with the aerodyne aerosol mass spectrometer, Mass Spectrometry Reviews, 26, 185-222, 10.1002/mas.20115, 2007.

Drewnick, F., Hings, S. S., Decarlo, P., Jayne, J. T., Gonin, M., Fuhrer, K., Weimer, S., Jimenez, J. L., Demerjian, K. L., and Borrmann, S.: A New Time-of-Flight Aerosol Mass Spectrometer (TOF-AMS)—Instrument Description and First Field Deployment, Aerosol Science & Technology, 39, 637-658, 2005.

Middlebrook, A. M., Bahreini, R., Jimenez, J. L., and Canagaratna, M. R.: Evaluation of Composition-Dependent Collection Efficiencies for the Aerodyne Aerosol Mass Spectrometer using Field Data, Aerosol Science and Technology, 46, 258-271, 10.1080/02786826.2011.620041, 2012.

- Was there source apportionment performed?

Reply: The source apportionment has been conducted and the details are given in the supplementary materials.
The potential source contribution function (PSCF) analysis method has been widely applied to study the potential source regions of pollutants (Ashbaugh et al., 1985; Wang, et al., 2009). The PSCF values for each grid cell (0.5°*0.5°) in the selected domain were calculated by counting the number of trajectories those terminated within each grid cell, as follows:

$$PSCF_{ij} = \frac{m_{ij}}{n_{ij}} \qquad (1)$$

where $n_{ij}$ is the number of endpoints that fall in the $ij^{th}$ cell, and $m_{ij}$ is the number of endpoints for the same cell with pollutant concentrations higher than the set criterion value. The PSCF values should be weighted according to $n_{ij}$. In this study, the weighting function ($W_{ij}$) was defined as follows:

$$W_{ij} = \begin{cases} 1.00 & 10 \times \overline{n_{ij}} < n_{ij} \\ 0.70 & 5 \times \overline{n_{ij}} < n_{ij} \leq 10 \times \overline{n_{ij}} \\ 0.40 & 2 \times \overline{n_{ij}} < n_{ij} \leq 5 \times \overline{n_{ij}} \\ 0.05 & n_{ij} \leq 2 \times \overline{n_{ij}} \end{cases} \qquad (2)$$

where $\overline{n_{ij}}$ is the mean $n_{ij}$ value. In this study, a potential source analysis was conducted for the nucleation and accumulation mode particles, which represented the air mass influence on the NPF event and the particles from long-range transport, respectively. The criterion values of $PM_{2.5}$ and $PM_{10}$ mass concentration were 75 µg cm$^{-3}$ and 100 µg cm$^{-3}$, which was the mean value in March, April and May in 2021.

The PSCF results (Fig. S7) showed that high $PM_{2.5}$ mass concentration at CAMS was dominated by two sources, the northwesterly and westerly originating air mass containing dust particles, and the southerly air mass with high mass loading of anthropogenic aerosols. However, for $PM_{10}$ mass concentration, the high values only contributed by the air masses passing through Inner Mongolia and carrying dust particles.

[Figure]

Fig. S7. Air mass classification of back trajectories arriving at the CAMS site in

March, April and May, 2021. The color bar indicates the number concentration weighted potential source contribution function (PSCF) value of (upper panel) $PM_{2.5}$ and (lower panel) $PM_{10}$ mass concentration.

Ashbaugh, L.L., Malm, W.C., Sadeh, W.Z., 1985. A residence time probability analysis of sulfur concentrations at Grand Canyon National Park. Atmospheric Environment 19 (8), 1263–1270.

- How were the dust events defined, not the distinction between the dust episodes, but the identification itself, was it based on airmass trajectories?

Reply: we give the definition in the section of 3.1 with original line 143-145. Three types of dust days are classified based on visibility (National Weather Bureau of China, 1979; Wang et al., 2005), including dust storm with visibility below 1.0 km; blowing dust with visibility of 1.0-10 km and floating dust with visibility below 10 km. The dust days we discussed in the manuscript were identified based on the above algorithm. In this study, the visibility data are from the national surface meteorological observation stations of China Meteorological Administration (CMA). Furthermore, the daily weather phenomena and visibility are issued by CMA (http://www.asdf-bj.net/publish/observation/5.html, last access on 23 March, 2023), which can also help to recognize the dust event. We conducted the identification of dust days during 2017-2021 spring based on the visibility data. We have supplemented the visibility data origins in the manuscript.

The 72 hours back trajectories on dust days March 15-17, 2021 were also calculated and given in the supplementary materials (Fig. S2). It showed the air mass passing through Mongolia, which is the source of dust storm.

---

## Author Response (AR2)

Response to reviewer 1

   Thanks for your great efforts and valuable comments, which helps to improve our manuscript. We have addressed the reviewers' comments on a point-to-point basis as below for consideration. Referee comments are in black. Author responses are in red.

The revised manuscript is greatly improved over the original one. However, there are still a few minor issues that need to be considered before I can recommend the paper to be accepted for publication:

The overall structure of section causes some confusion, which should be clarified. While section 3.1 discusses all the observed dust events, sections 3.2-3.5 evidently concentrated on a single dust event. This is not clear when reader the paper for the first time 1) because only the title of section 3.2 refers to a case study and sections 3.3-3.5 do not, and 2) because the contents of sections 3.3-3.5 have very little information on to which dust event(s) the data in them refers to (except a few dates without a year here and there). I recommend combining these sections into a single one (e.g. 3.2 Case study of a dust storm), and sections 3.3-3.5 put under the same title as sub-sections 3.2.1-3.2.3.
Reply: Thanks for your comment. We re-organized the structure of this manuscript. The original section 3.2 and 3.5 was combined into section 3.2, and this section was sub-divided into the following parts:
3.2 Case study of a dust-related NPF event
3.2.1 A severe dust storm case
3.2.2 Secondary aerosol formation during dust storm
3.2.3 Variations of particle hygroscopicity
3.2.4 Impact on the cloud condensation nuclei by the dust storm

The authors say that they have removed their statements on the role of anthropogenic emission from abstract and discussion. However, there are still claims in section 3.1 that are not solid in this respect (e.g. lines 235-236, lines 241-244). Please check out and revised if necessary.
Reply: As compared with the original submitted manuscript, we removed the statements of the quantitative description of the influence of anthropogenic emissions on NPF events from the abstract and conclusions, as this evaluation method is not scientifically sound as the reviewer suggested. This kind of statements was probably not robust, which could not be presented in the abstract or conclusion part. However, we still believe the influence of anthropogenic emissions on dust-related NPF events could be different from that on the normal NPF events. For this purpose, we prefer to have some discussions about anthropogenic emission effect on NPF in this section. In line 235-236, it was revised to "$N_{3-10}$ and $N_{3-25}$ were generally lower on dust-related NPF days than on other NPF days, which was probably due to a considerable contribution by anthropogenic emissions on non-dust NPF days."

For line 241-244, we added a reference to support our discussion. The sentence has been revised to "This suggested that the influence of anthropogenic emitted precursors on non-dust days when nucleated particles growing into the sizes above 10 nm could be more significant. It has been also reported the nitrogen-containing oxygenated organic molecules related with anthropogenic emissions in urban Beijing can contribute over 50% to the particle growth (Qiao et al., 2021). However, the influence by anthropogenic emissions was difficult to be estimated, as even during the growth process of dust-related NPF events, freshly-emitted precursors could also participate."

Qiao, X., Yan, C., Li, X., Guo, Y., Yin, R., Deng, C., Li, C., Nie, W., Wang, M., Cai, R., Huang, D., Wang, Z., Yao, L., Worsnop, D. R., Bianchi, F., Liu, Y., Donahue, N. M., Kulmala, M. and Jiang, J.: Contribution of Atmospheric Oxygenated Organic Compounds to Particle Growth in an Urban Environment, Environmental science & technology, 55(20): 13646-13656, DOI: 10.1021/acs.est.1c02095, 2021.

Line 172: should this be vis > 10 km?
Reply: Even on floating dust days, the visibility should be below 10 km, according the dust case identification method suggested by Wang et al., (2005). Floating dust is the weakest as compared with dust storm and blowing dust, and is generally characterized with fine dust particles suspending in the lower troposphere with horizontal visibility of o10,000 m.

Wang, S., Wang, J., Zhou, Z. and Shang, K.: Regional characteristics of three kinds of dust storm events in China, Atmospheric Environment, 39(3): 509-520, DOI: 10.1016/j.atmosenv.2004.09.033, 2005.

Lines 193-194: this is a bit strange statement. I suppose you mean that NPF was not observed until the dust event was over (now you kind of claim contrary to this).
Reply: The sentence has been corrected to "NPF event can not be observed until the whole dust process finished."

Lines 225-227: I do not feel that referring to fractions when comparing particle growth rates between the two types of days is a proper term here. Maybe it would be better to talk about ratios.

Reply: this sentence has been corrected to "The ratio of $GR_{dust\_NPF}$ to $GR_{other\_NPF}$ ranged from 0.50 to 0.86, with a mean value of approximately 0.67."

Response to reviewer 2

Thanks for your great efforts and valuable comments, which helps to improve our manuscript. We have addressed the reviewers' comments on a point-to-point basis as below for consideration. Referee comments are in black. Author responses are in red. All corrections have been conducted according to reviewer's comments in the manuscript and supplementary materials.

While the authors addressed several of the comments raised by the reviewers in the first round, I believe there is still room for improvement and some claims that are either false or unjustified throughout the manuscript that need to addressed prior to publication in ACP. I highlight the major points here:

1. The authors consider dusty days are the 'background' conditions of the atmosphere in Beijing given the low average CS compared to other NPF-event day. In practice, dust events are extreme events and cannot be considered a norm or background. Instead, Beijing is recurrently subject to clean air-masses arriving from the north and west, not carrying any dust, but are low in CS. One can note based on the violin plots in Figure 2, that there are data points in the 'other NPF' category which are low in CS. Therefore, for an improved understanding of the dust episodes, and other episodes, a comparison between NPF characteristics (CS, J, GR) on 'dust days' and 'clean-other days' and 'polluted-other days' can be added.

Reply: The hourly mean $CS$ values ranged from 0.002 to 0.163 $s^{-1}$ during our study period, with statistical mean, median, 25% and 75% value of 0.034, 0.027, 0.014 and 0.047 $s^{-1}$, respectively. The potential source contribution function (PSCF) analysis was conducted based on the back trajectories calculation, with the $CS$ criterion value of 0.027 $s^{-1}$, which was the median $CS$ value during the measurement and the conditions with $CS$ above this value was regarded as the polluted conditions. The PSCF result has also revealed that the higher $CS$ values usually corresponded to the southerly regional air mass, containing high concentration of anthropogenic pollutants. The violin plot of $CS$ in the manuscript has been revised to be Fig. S2. The comparison of formation rate and growth rate was also supplemented as shown in Fig. S3.

[Figure]

Fig. S1. Air mass classification of back trajectories arriving at the CAMS site in March, April and May, 2021. The color bar indicates the number concentration

weighted potential source contribution function (PSCF) value of condensation sink.

[Figure]

Fig. S2 The violin plot of condensation sink (*CS*) of dust-related NPF (Dust_NPF) and other NPF events under clean (Clean NPF) and polluted conditions (Polluted NPF). The marker represents the median value; a box indicating the interquartile range, and the shaded area represents the distribution probability of the *CS*.

[Figure]

Fig. S3 The violin plot of formation rate, $J_{obs}$ (a) and growth rate, GR (b) of dust-related NPF (Dust_NPF) and other NPF events under clean (Clean NPF) and polluted conditions (Polluted NPF). The marker represents the median value; a box indicating the interquartile range, and the shaded area represents the distribution probability of $J_{obs}$ and GR.

2. Sentence on line 194 reads: 'On May 7 and May 8, NPF events were observed with extremely low CS values of approximately 0.0025 – 0.003 s$^{-1}$, indicating the concentration level of precursors participating nucleation and growth were comparable for these two cases.' In this sentence, the authors claim that they are able to 'estimate' the precursor concentration based on the CS level. This is an incorrect way of addressing atmospheric observations as the emissions, oxidants and meteorology are ignored. It is important to note here that May 7, 2021 is a workday, while May 8, 2021 is a weekend, which means that the emissions are definitely not comparable. See for example: https://doi.org/10.1007/s11430-008-0088-2. Here, the authors could check the changes in SO2 which is a precursor of the main vapor driving nucleation in Beijing

(sulfuric acid).

Reply: The authors all agreed with the reviewer's comment that we should look into the reactive gases, condensation sink (*CS*), and sulfuric acid data on May 7 and 8 further. The reactive gases ($SO_2$, $NO_2$, and $O_3$) were derived as the average of the values at four air quality monitoring sites, including Guanyuan (GY), Wanshou Temple (WST), Dongsi (DS), and Chaoyang (CY), in urban Beijing, as mentioned in the manuscript. It showed before NPF start, around 8:00 LT, the concentration of $SO_2$ and $NO_2$ on May 7 and 8 was quite close, which did not show a clear difference between workday and weekend. Furthermore, the *CS* before NPF start was also comparable on these two days, around 0.003 s$^{-1}$, indicating the available condensable vapor was close.

[Figure]

Fig. S4. Time series of hourly volume mixing ratio of $SO_2$, $NO_2$, $O_3$ and condensation sink (*CS*) on May 7 and 8, 2021.

As there is no direct $H_2SO_4$ measurement data available in this work, we used two methods to estimate sulfuric acid concentration. In the calculation of [$H_2SO_4$] in Beijing, we chose proxy equation number 2 as Proxy 1 in this study (Eq. 1) and 7 (Eq. 2) as Proxy 2 in this study as recommended by Lu et al. (2019), to represent the simplest and most accurate method, respectively.

$$[H_2SO_4] = 280.05 \times UVB^{0.14} \times [SO_2]^{0.40} \qquad (1)$$
$$[H_2SO_4] = 0.0013 \times UVB^{0.13} \times [SO_2]^{0.40} \times CS^{-0.17} \times ([O_3]^{0.44} + [NO_x]^{0.41}) \quad (2)$$

And the UVB was derived by 0.008% × Glob_R, based on the previous study that the monthly average of the ratio of UVB to global radiation (Glob_R) ranged from 0.007

to 0.017% in Beijing (Hu et al., 2013). The average ratio of January and February (0.008%) was applied.

[Figure]

Fig. S4. The sulfuric acid concentrations derived by different proxy equations. The blue and orange lines indicate the result by N2 (Proxy 1) and N7 (Proxy 2) method by Lu et al., 2019

Hu, B., Zhang, X. H. and Wang, Y. S.: Variability in UVB radiation in Beijing, China, Photochem Photobiol, 89(3): 745-750, DOI: 10.1111/php.12051, 2013.

Lu, Y., Yan, C., Fu, Y., Chen, Y., Liu, Y., Yang, G., Wang, Y., Bianchi, F., Chu, B., Zhou, Y., Yin, R., Baalbaki, R., Garmash, O., Deng, C., Wang, W., Liu, Y., Petäjä, T., Kerminen, V. M., Jiang, J., Kulmala, M. and Wang, L.: A proxy for atmospheric daytime gaseous sulfuric acid concentration in urban Beijing, Atmos. Chem. Phys., 19(3): 1971-1983, DOI: 10.5194/acp-19-1971-2019, 2019.

Tang, W., Zhao, C., Geng, F., Peng, L., Zhou, G., Gao, W., Xu, J. and Tie, X.: Study of ozone "weekend effect" in Shanghai, Science in China Series D: Earth Sciences, 51(9): 1354-1360, DOI: 10.1007/s11430-008-0088-2, 2008.

3. An in depth analysis of the particle formation and growth rates as well as CS can still be performed, for instance is there a difference between particle formation rates on the different types of dust events? Or was the formation rate on the severe dust storm higher than the rest? The same applies to the growth rates and condensation sink. A plot of J vs GR/CS can also be useful here to show where the dust points fall compared to others. Reply: As the reviewer recommended, we supplemented a scatterplot of $J_{obs}$, $GR$ and $CS$ and also categorized by different NPF event types, which included the NPF events occurred under clean and polluted conditions and influenced by blowing dust (BD), floating dust (FD) and dust storm (DS). There is no clear relationship between $J_{obs}$ and $GR$ as shown in Fig. S5. $J_{obs}$ ranged from 0.3 to 23.6 cm$^{-3}$s$^{-1}$, with the mean value of 6.1 cm$^{-3}$s$^{-1}$. $GR$ ranged from 1.1 to 8.9 nm h$^{-1}$, with the mean value of 4.18 nm h$^{-1}$. The mean $J_{obs}$ of NPF events under clean (96), polluted (10), BD (9) and FD (11) conditions was 5.7, 8.7, 6.6 and 6.8 cm$^{-3}$s$^{-1}$, respectively. The corresponding mean $GR$ value was 4.2, 4.3 4.1 and 3.7 nm h$^{-1}$, respectively, and mean $CS$ was 0.006, 0.020, 0.005 and

0.005 s$^{-1}$. Formation rate under polluted conditions was significantly higher than those under other conditions, suggesting there were abundant condensing vapours participating nucleation and overcame the competition with the pre-existing particles (as indicated by high $CS$ value). The $J_{obs}$ at 3 nm and $GR$ reported in this study was comparable with the values of previous studies, 0.5-20 cm$^{-3}$s$^{-1}$ and a few nm h$^{-1}$ to 20 nm h$^{-1}$, respectively, as summarized by Chu et al. (2019) based on several NPF studies in China. However, due to the limited cases of NPF with moderate accurate $J_{obs}$ and $GR$ influenced by BD (number of cases = 9), FD (11) and DS (1), a confident comparison results among different dust NPF events could not be derived. The related figure and discussion have been supplemented in the manuscript.

[Figure]

Fig. S5, Scatter plot of formation rate ($J_{obs}$), grow rate (GR) and condensation sink (CS) as categorized by different NPF event types, including the cases occurring under clean, polluted conditions and influenced by blowing dust (BD), floating dust (FD) and dust storm (DS).

4. I am also surprised that the authors do not compare their results to any other study in Beijing, or China or worldwide. For example how does the J, GR and CS observed during those measurements compare to others? The work already done on new particle formation in Beijing is comprehensive and could be useful for the authors to improve their story, especially the part related to anthropogenic emissions.

Reply: Thanks for the reviewer's suggestion. The NPF studies have been conducted extensively since 1990 worldwide and 2000 over China. Kerminen et al., (2018) and Chu et al., (2019) have summarized the particle nucleation and growth based on filed campaigns worldwide and over China, respectively. In this work, we only focused on the NPF event in spring time. In the previous studies, the long-term datasets are limited,

especially in China. The formation and growth rates showed clear seasonal variation, and also varied depending on the environments. The lower limit of the particle size distribution should also be considered, as it influences the formation rate calculation. So, we focused on the comparison between this wok and the previous work conducted in Beijing in spring time or at least above 1 year with the particle detection limit of 3 nm. Based on the one-year study of NPF events at Peking University (PKU) site in Beijing in 2004, it has been reported that the formation rate ($J_3$) ranged from 3.3 to 81.4 $cm^{-3}s^{-1}$, and growth rate (GR) ranged from 0.1 to 11.2 nm $h^{-1}$, respectively (Wu et al., 2007). Wang et al. (2013) has also reported $J_3$ at PKU site ranged from 2.2 to 34.5 $cm^{-3}s^{-1}$, and growth rate ranged from 2.5 to 15.3 nm $h^{-1}$ from March to November in 2008. PKU site locates 5 km to the north of CAMS site, which is a representative urban site in Beijing with long-term study of NPF events. Based on the long-term study at PKU site (2013-2019), it has been recently reported the annual average of $J_3$ decreased from 12 $cm^{-3}s^{-1}$ in 2013 to 3 $cm^{-3}s^{-1}$ in 2017, whereas increased to 5 $cm^{-3}s^{-1}$ in 2019, and GR values kept stable around 2-4 nm $h^{-1}$ during these years (Shang et al., 2022). The mean values of $J_3$ and GR in our study was 6.10 $cm^{-3}s^{-1}$ and 4.18 nm $h^{-1}$, which was comparable with the values reported by Shang et al., (2022).

Although the precursors from anthropogenic emissions have been proved to participating the particle nucleation and growth processes, it is difficult to quantify its contribution, especially in megacities like Beijing (Kulmala et al., 2021). The complex primary emissions, for example, traffic emissions with plentiful nanoparticles, can mix with the freshly nucleated particles, making it difficult to resolve the particles from primary emissions and secondary formation. However, based some long-term studies of NPF event, it has reported that the decrease of the precursors due to the emission control strategies in China has caused formation rate reduction from 2013 to 2017 both in Beijing (Shang et al., 2022) and rural site in Yangtze River Delta region (Shen et al., 2022).

The comparison between this work and the previous studies has been added in the manuscript.

Wu, Z., Hu, M., Liu, S., Wehner, B., Bauer, S., Ma ßling, A., Wiedensohler, A., Petäjä, T., Dal Maso, M. and Kulmala, M.: New particle formation in Beijing, China: Statistical analysis of a 1-year data set, Journal of Geophysical Research, 112(D9), DOI: 10.1029/2006jd007406, 2007.

Chu, B., Kerminen, V.-M., Bianchi, F., Yan, C., Petäjä, T. and Kulmala, M.: Atmospheric new particle formation in China, Atmospheric Chemistry and Physics, 19(1): 115-138, DOI: 10.5194/acp-19-115-2019, 2019.

Kerminen, V.-M., Chen, X., Vakkari, V., Petäjä, T., Kulmala, M. and Bianchi, F.: Atmospheric new particle formation and growth: review of field observations, Environ. Res.Lett., 13(10), DOI: 10.1088/1748-9326/aadf3c, 2018.

Shang, D., Tang, L., Fang, X., Wang, L., Yang, S., Wu, Z., Chen, S., Li, X., Zeng, L., Guo, S. and Hu, M.: Variations in source contributions of particle number concentration under long-term emission control in winter of urban Beijing, Environ Pollut, 304: 119072, DOI: 10.1016/j.envpol.2022.119072, 2022.

Kulmala, M., Dada, L., Daellenbach, K. R., Yan, C., Stolzenburg, D., Kontkanen, J., Ezhova, E., Hakala, S., Tuovinen, S., Kokkonen, T. V., Kurppa, M., Cai, R., Zhou, Y., Yin, R., Baalbaki, R., Chan, T., Chu, B., Deng, C., Fu, Y., Ge, M., He, H., Heikkinen, L., Junninen, H., Liu, Y., Lu, Y., Nie, W., Rusanen, A., Vakkari, V., Wang, Y., Yang, G., Yao, L., Zheng, J., Kujansuu, J., Kangasluoma, J., Petaja, T., Paasonen, P., Jarvi, L., Worsnop, D., Ding, A., Liu, Y., Wang, L., Jiang, J., Bianchi, F. and Kerminen, V. M.: Is reducing new particle formation a plausible solution to mitigate particulate air pollution in Beijing and other Chinese megacities?, Faraday Discuss, 226: 334-347, DOI: 10.1039/d0fd00078g, 2021.

Shen, X., Sun, J., Ma, Q., Zhang, Y., Zhong, J., Yue, Y., Xia, C., Hu, X., Zhang, S. and Zhang, X.: Long-term trend of new particle formation events in the Yangtze River Delta, China and its influencing factors: 7-year dataset analysis, Science of the Total Environment: 150783, DOI: https://doi.org/10.1016/j.scitotenv.2021.150783, 2022.

5. The same applies to all other results in the paper. The authors do not acknowledge the work of previous colleagues who measured long-term aerosol mass composition in China. How do the results in Figure 8 compare to those studies? The same applies for the hygroscopicity analysis and the SOR/NOR results. Where do these results stand in comparison to other literature?

Reply: The chemical composition and hygroscopicity analysis was only conducted for the dust storm study from March 15-16, 2021, due to the limited dataset. In the previous studies in Beijing, it has been reported the NR-PM$_1$ derived from AMS could reach ~200 µg m$^{-3}$ during polluted episode and decreased to several µg m$^{-3}$ under clean conditions (Zhang et al., 2018). In this work, the PM$_1$ mass concentration ranged from approximately 5.0 µg m$^{-3}$ during dust and post dust period and to 83.2 µg m$^{-3}$ during a moderate polluted conditions before dust. During the dust and post dust period, organics was the dominant contributor to the chemical composition, which was consistent with the previous studies that organics could contributed 40-60% to PM$_1$ in Beijing (Zhang et al., 2018; 2019). However, the mass fraction of nitrate and ammonium increased during polluted condition before dust, which has been also reported a recent study in Beijing in January - February, 2021 (Zhang et al., 2023). The limited sample of chemical composition and hygroscopicity measurement could introduce uncertainties in the comparison between the results of this study with other previous work.

The hygroscopicity parameter ($\kappa$) of 50 nm ranged from 0.05 to 0.17, and 0.15-0.30 for 100 nm particles, which was consistent with the long-term study in Beijing as reported by Wang et al. (2018) and Zhang et al. (2023). Although the chemical composition of ultrafine particles can not be derived in this work, based on a recent study in Beijing, it has been revealed that organics dominated the mass concentration of particles below 100 nm, whereas the mass fraction of nitrate increased depending on the size (Li et al., 2023).

The SOR and NOR results have been compared with the previous studies in Beijing. It has been reported that the SOR was 0.18 in clean air conditions, whereas it was 0.27 under polluted conditions in Beijing in 2016 wintertime, indicating that SO2 secondary transformation was a major pathway of sulfate production with a higher conversion

efficiency under the polluted episode, whereas NOR was approximately 0.08, under both clean and polluted conditions (Wu et al., 2019).

The discussion of chemical composition and hygroscopicity have been supplemented in the manuscript.

Li, X., Chen, Y., Li, Y., Cai, R., Li, Y., Deng, C., Yan, C., Cheng, H., Liu, Y., Kulmala, M., Hao, J., Smith, J. N. and Jiang, J.: Seasonal variations in composition and sources of atmospheric ultrafine particles in urban Beijing based on near-continuous measurements, Atmos. Chem. Phys. Diss., DOI: 10.5194/egusphere-2023-809, 2023.

Wang, Y., Wu, Z., Ma, N., Wu, Y., Zeng, L., Zhao, C. and Wiedensohler, A.: Statistical analysis and parameterization of the hygroscopic growth of the sub-micrometer urban background aerosol in Beijing, Atmospheric Environment, 175: 184-191, DOI: 10.1016/j.atmosenv.2017.12.003, 2018.

Zhang, Y., Wang, Y., Zhang, X., Shen, X., Sun, J., Wu, L., Zhang, Z. and Che, H.: Chemical Components, Variation, and Source Identification of PM1 during the Heavy Air Pollution Episodes in Beijing in December 2016, Journal of Meteorological Research, 32(1): 1-13, DOI: 10.1007/s13351-018-7051-8, 2018.

Zhang, Y., Vu, T. V., Sun, J., He, J., Shen, X., Lin, W., Zhang, X., Zhong, J., Gao, W., Wang, Y., Fu, T. M., Ma, Y., Li, W. and Shi, Z.: Significant Changes in Chemistry of Fine Particles in Wintertime Beijing from 2007 to 2017: Impact of Clean Air Actions, Environ Sci Technol, 54(3): 1344-1352, DOI: 10.1021/acs.est.9b04678, 2019.

Zhang, Y., Tian, J., Wang, Q., Qi, L., Manousakas, M. I., Han, Y., Ran, W., Sun, Y., Liu, H., Zhang, R., Wu, Y., Cui, T., Daellenbach, K. R., Slowik, J. G., Prévôt, A. S. H. and Cao, J.: High-time-resolution chemical composition and source apportionment of $PM_{2.5}$ in northern Chinese cities: implications for policy, Atmos. Chem. Phys. Diss., DOI: 10.5194/egusphere-2023-457, 2023.

Zhang, S., Shen, X., Sun, J., Che, H., Zhang, Y., Liu, Q., Xia, C., Hu, X., Zhong, J., Wang, J., Liu, S., Lu, J., Yu, A. and Zhang, X.: Seasonal variation of particle hygroscopicity and its impact on cloud-condensation nucleus activation in the Beijing urban area, Atmospheric Environment, 302, DOI: 10.1016/j.atmosenv.2023.119728, 2023.

6. Figure 4 is not readable, the dust and other NPF $N_{25}$ line colors are the very similar. The comparison to other locations, intercomparison between dust event types could be added.

Reply: We have removed figure 4 and 5 in the manuscript and supplemented a modified figure describing the mean diurnal variation of number concentration of particles in the size range of 3-25 nm ($N_{3-25}$) and geometric mean diameter ($D_{p,g}$) of NPF events occurred under clean, polluted conditions and influenced by blowing dust (BD), floating dust (FD) and dust storm (DS), respectively. $N_{3-25}$ showed similar diurnal pattern, which peaked around noon time governed by NPF event. The lower $N_{3-25}$ was found on DS-related NPF event, with lower $D_{p,g}$ at the initial growth stage below 20 nm, indicating less precursors participating nucleation and growth processes. It should

be also addressed that only one DS-related NPF event occurred (March 16, 2021) in this study, which could not represent overall characteristics of NPF events influenced by dust storm. $D_{p,g}$ on the dust-related NPF events (including BD, FD and DS type) in Fig. 5b was generally lower than that on the clean and polluted NPF events with nucleated particles below 20 nm. A quick growth of nucleated particles at round 19-20 LT was probably associated with the wind direction change as given in the supplementary materials (Fig. S9). The previous studies in Beijing have revealed that the southerly air masses containing plentiful anthropogenic precursors, facilitating the nucleation and growth processes of NPF events (Wang et al., 2013; Shen et al., 2018). It was also found polluted-NPF events usually started later, at around 10 LT, with $N_{3-25}$ quickly peaked at around 12 LT, indicating a shorter nucleation process with higher formation rate as shown above.

[Figure]

Fig. S6. the mean diurnal variation of number concentration of particles in the size range of 3-25 nm ($N_{3-25}$) (a) and geometric mean diameter ($D_{p,g}$) (b) of NPF events occurred under clean, polluted conditions and influenced by blowing dust (BD), floating dust (FD) and dust storm (DS), respectively.

7. There are several claims that remain in the text, not justified by the observation nor backed-up by previous literature. I give some examples here, but the entire manuscript benefits from being revised:

Reply: We have revised the discussions with not robust confidence as the reviewer mentioned below and also checked through all the manuscript.

i) 'The variation in $NO_2$ sharply increased in the early morning (5:00–6:00 LT) on March 15, which could be attributed to the downward mixing of $NO_2$ rich air in the residual layer where $NO_2$ was trapped during the pollution episode on March 14.' Do the authors have proof of this? Neither a citation of a previous similar observation, nor a discussion is added.

Reply: We looked into the time series of wind distribution depending on air pressure level (800-1000 hpa) on March 14th and 15th based on the reanalysis meteorological data (https://cds.climate.copernicus.eu/). The wind (u- and v-component) data for the Beijing region (latitude:39-42°N, longitude: 115-117°E) was averaged and given in Fig. S1. Before March 15, the air pollutants accumulated in Beijing, with stable upward wind. The wind changed on 3:00 LT March 15, with the wind direction from upward to downward. Although quite small wind below 900 hpa (corresponding to below ~1000 m geopotential height) near ground surface was observed before 06:00 on March 15, the wind direction switched to downward at around 3:00 LT and could bring the pollutants to the ground, resulting in the elevated $NO_2$ and condensation sink as given in Fig. 8 in the manuscript. The air pollutants decreased sharply after 6:00 LT as the wind speed increased. We have revised the discussion in the manuscript and the figure was added in the supplementary materials.

[Figure]

Fig. S7. Time series of wind variation depending on the air pressure level from March 14 to 15, the arrows represent the wind direction and the length of the arrow indicates the wind speed (m/s)

ii) 'The concentration of $SO_2$ remained stable before 6:00 LT, probably because its distribution was uniform in the boundary layer', same as the previous sentence.
Reply: As illustrated above, the wind field remained stable before 3:00 on March 15 when the air pollutants accumulated. The wind direction switched from upward to downward since 3:00 LT and the wind speed increased significantly since 6:00 LT. The mixing ratio of $SO_2$ remained stable decreased sharply until 6:00 LT due to the strong wind, indicating the vertical distribution of $SO_2$ was uniform in the boundary layer.

iii) 'The concentration of $O_3$ increased during dust storms, probably because the $O_3$ budget was influenced by mineral dust.' Can the authors prove this based on observations or previous literature?
Reply: It has been revised to "The volume mixing ratio of $NO_2$ decreased, while that of $O_3$ increased, indicating that the removal of $NO_2$ was helpful for the elevated $O_3$ concentration, as $NO_x$-titration photochemistry process could influence the production and loss of $O_3$ (Lu et al., 2010). It has been also reported by the previous study in

Beijing-Tianjin-Hebei region, the decrease in $NO_x$ increased ozone and enhanced the atmospheric oxidizing capacity (Huang et al., 2020)."

Huang, X., Ding, A., Gao, J., B. Zheng, D. Zhou, X. Qi, R. Tang, J. Wang, C. Ren, W. Nie, X. Chi, Z. Xu, L. Chen, Y. Li, F. Che, N. Pang, H. Wang, D. Tong, W. Qin, W. Cheng, W. Liu, Q. Fu, B. Liu, F. Chai, S.J. Davis, Q. Zhang and K. He. Enhanced secondary pollution offset reduction of primary emissions during COVID-19 lockdown in China, National Sci Rev, nwaa137, 2020.

Lu, K., Zhang, Y., Su, H., Brauers, T., Chou, C. C., Hofzumahaus, A., Liu, S. C., Kita, K., Kondo, Y., Shao, M., Wahner, A., Wang, J., Wang, X. and Zhu, T.: Oxidant (O3 +NO2) production processes and formation regimes in Beijing, J. Geophys. Res., 115: D07303, DOI: 10.1029/2009JD012714, 2010.